# Huntingtin recruits KIF1A to transport synaptic vesicle precursors along the mouse axon to support synaptic transmission and motor skill learning

**Hélène Vitet**[1†, ‡], **Julie Bruyère**[1], **Hao Xu**[2], **Claire Séris**[1], **Jacques Brocard**[1§], **Yah-Sé Abada**[3], **Benoît Delatour**[3], **Chiara Scaramuzzino**[1*], **Laurent Venance**[2], **Frédéric Saudou**[1*#]

[1]Univ. Grenoble Alpes, Inserm, U1216, CHU Grenoble Alpes, Grenoble Institut Neuroscience, Grenoble, France; [2]Center for Interdisciplinary Research in Biology, College de France, CNRS, INSERM, Université PSL, Paris, France; [3]Sorbonne Université, Institut du Cerveau, Paris Brain Institute, ICM, Inserm U1127, CNRS UMR7225, Paris, France

**\*For correspondence:**
chiara.scaramuzzino@univ-grenoble-alpes.fr (CS);
frederic.saudou@inserm.fr (FS)

**Present address:** [†]Department of Pediatrics, College of Medicine, National Cheng Kung University, Tainan City, Taiwan; [‡]Graduate Institute of Mind, Brain and Consciousness (GIMBC), Taipei Medical University, Taipei, Taiwan; [§]PLATIM, SFR Biosciences (UAR3444/CNRS, US8/Inserm, ENS de Lyon, UCBL), Lyon, France

[#]Lead contact

**Competing interest:** The authors declare that no competing interests exist.

**Abstract** Neurotransmitters are released at synapses by synaptic vesicles (SVs), which originate from SV precursors (SVPs) that have traveled along the axon. Because each synapse maintains a pool of SVs, only a small fraction of which are released, it has been thought that axonal transport of SVPs does not affect synaptic function. Here, studying the corticostriatal network both in microfluidic devices and in mice, we find that phosphorylation of the Huntingtin protein (HTT) increases axonal transport of SVPs and synaptic glutamate release by recruiting the kinesin motor KIF1A. In mice, constitutive HTT phosphorylation causes SV over-accumulation at synapses, increases the probability of SV release, and impairs motor skill learning on the rotating rod. Silencing KIF1A in these mice restored SV transport and motor skill learning to wild-type levels. Axonal SVP transport within the corticostriatal network thus influences synaptic plasticity and motor skill learning.

## Editor's evaluation

This important study provides solid in vitro and in vivo data supporting the connection between Huntingtin's (HTT) phosphorylation state and the recruitment of Kif1A in the axonal anterograde trafficking of synaptic vesicles precursors. This works highlights the role of HTT-Kif1A synergy in modulating synaptic vesicle precursor transport and synaptic function, adding important information related to how synaptic vesicle pools are formed and replenished.

## Introduction

Neurons communicate by transmitting chemical messages through their synapses. The number of synaptic vesicles (SVs) that carry these chemical messengers (neurotransmitters), the probability of the vesicles releasing neurotransmitter, and the vesicle quantal size, all affect synaptic strength and thus the ability to learn and remember (*Katz, 1969*). SVs actually begin life as SV precursors (SVPs), which are formed in the cell body and transported along the axon to the presynapse—a distance that can span meters (*Guedes-Dias and Holzbaur, 2019*; *Rizalar et al., 2021*; *Rizzoli, 2014*). One would expect that this long-distance axonal transport should influence SV homeostasis, because synapses must somehow be replenished with new vesicles as they release neurotransmitters. Thus far, however,

evidence for this intuition has been lacking. It is thought that the synaptic SV pools, which contain hundreds of vesicles of which only a few percent participate in synaptic release (*Reshetniak and Rizzoli, 2021*), are sufficient to ensure a ready supply of SVs to be released, even with prolonged neuronal stimulation (*Denker et al., 2011*; *Rizzoli, 2014*). If necessary, neighboring synapses can draw on local SV pools that circulate between them (*Wong et al., 2012*).

Despite this substantial reserve of SVs, mutations that strongly affect SVP transport have been found to affect neuronal transmission and behavior in mice, flies, and worms. In *Caenorhabditis elegans*, null mutants for the kinesin-related gene *unc-104* or the vesicle-associated protein SAM-4 lead to defects in SVP transport, with a consequent lack of SVs at synapses and locomotor deficits (*Hall and Hedgecock, 1991*; *Zheng et al., 2014*). In *Drosophila*, deletion of the *imac* gene, a kinesin-3 family member, impairs SVP axonal transport and the formation of synaptic boutons (*Pack-Chung et al., 2007*). In mice, loss of function of *unc-104*'s mammalian homologue, KIF1A (*Okada et al., 1995*), leads to the accumulation of SVPs in the cell body and a dramatic reduction in the number of SVs at synapses, along with sensorimotor deficits and early postnatal death (*Yonekawa et al., 1998*). Completely blocking a molecular motor, however, does not tell us whether more modest enhancement or attenuation of axonal transport influences synapse homeostasis, synaptic transmission, or the function of specific brain circuits.

Mice bearing mutations in Huntingtin (HTT), a protein that plays a prominent role in axonal transport (*Saudou and Humbert, 2016*), provide one model system that could illuminate the role of SVP transport on synaptic homeostasis. HTT scaffolds various cargoes—endosomes, autophagosomes, vesicles containing BDNF, APP, etc. (*Bruyère et al., 2020*; *Cason et al., 2021*; *Caviston et al., 2011*; *Fu and Holzbaur, 2014*; *Gauthier et al., 2004*; *Her and Goldstein, 2008*; *Liot et al., 2013*; *Wong and Holzbaur, 2014*)—along with the appropriate molecular motors for anterograde or retrograde transport and any adaptor proteins that may be needed. Because the direction of HTT-mediated transport is dictated by its phosphorylation at serine 421 (*Bruyère et al., 2020*; *Colin et al., 2008*; *Ehinger et al., 2020*; *Vitet et al., 2020*), we were able to investigate how mutations at this site affect axonal transport of SVPs. First we studied SVP transport in a reconstituted neuronal circuit on-a-chip, then in mice that bear either a constitutively phosphorylated HTT mutant (S421D) or an unphosphorylatable one (S421A). Our data reveal a functional link between anterograde transport of SVPs within corticostriatal projecting neurons, the synaptic SV pools, and the release probability of SVs at corticostriatal synapses, with consequences for motor skill learning.

## Results

### Constitutive HTT phosphorylation increases anterograde SVP transport and synaptic glutamate release

To investigate whether HTT and its phosphorylation affect the transport of SVP in a physiologically relevant system, we reconstituted mature corticostriatal circuits in microfluidic devices (*Moutaux et al., 2018*; *Virlogeux et al., 2018*). These devices have been used to establish HTT's role in the transport of organelles such as BDNF-containing vesicles and signaling endosomes (*Gauthier et al., 2004*; *Liot et al., 2013*; *Scaramuzzino et al., 2022*; *Virlogeux et al., 2018*). Microfluidics consist of a presynaptic and a postsynaptic compartment containing cortical and striatal neurons, respectively, and a middle synaptic compartment that receives axons from cortical neurons and dendrites originating from striatal neurons (*Figure 1A*). The three compartments are connected by 3-μm-wide microchannels that are 500 μm long for axons and 75 μm long for dendrites (*Lenoir et al., 2021*). The number of striatal axons reaching the synaptic chamber at maturity is limited by the generation of a laminin gradient from the cortical chamber to the striatal chamber, whilst the poly-D-lysine concentration is kept constant (*Scaramuzzino et al., 2022*). In this configuration, isolated cortical axons unilaterally connect postsynaptic striatal dendrites in the middle compartment enriched in functional synaptic contacts (*Ehinger et al., 2020*; *Moutaux et al., 2018*; *Scaramuzzino et al., 2022*; *Virlogeux et al., 2018*).

We generated cortical and striatal neurons from embryos of wild-type (WT) mice and mice bearing either constitutively phosphorylated (HTT-SD mice) or unphosphorylatable HTT (HTT-SA mice) at embryonic day 15.5 (E15.5). HTT phosphorylation at serine 421 is mimicked by the replacement of the serine by an aspartic acid, which maintains the positive charge (S421D), whereas the

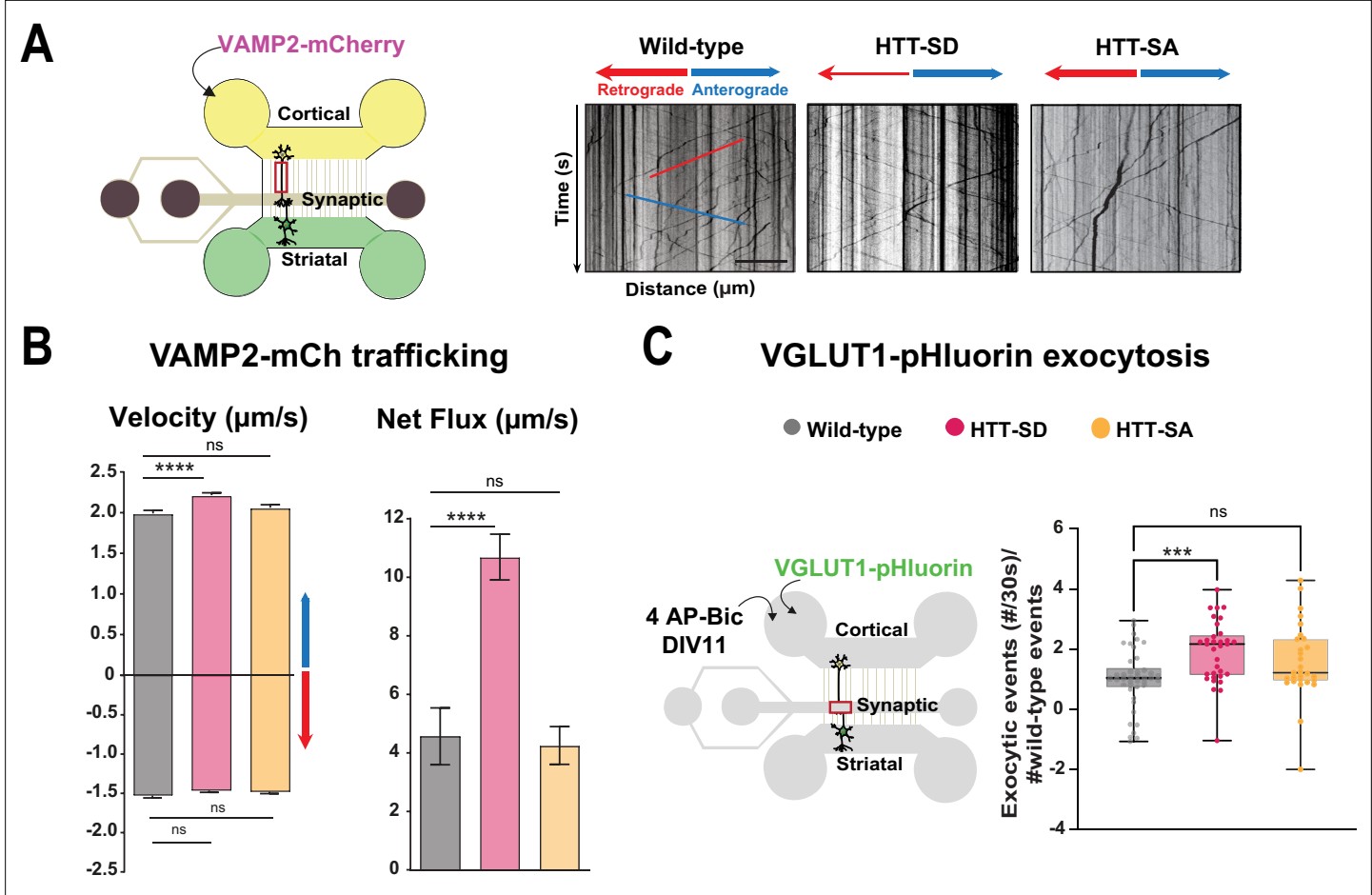

**Figure 1.** HTT phosphorylation at S421 increases synaptic vesicle precursor (SVP) anterograde axonal transport and SV exocytosis. (**A**) Diagram of the microfluidic device for reconstituting a corticostriatal network compatible with live-cell imaging of axons. Cortical axons grow in the cortical chamber (yellow) and connect with the striatal dendrites in the striatal chamber (green) through synapses in the synaptic compartment (purple). On the right, representative kymographs of VAMP2-mCherry vesicle transport in axons for each genotype. Scale bar = 25 µm. (**B**) Segmental anterograde (**** p<0.0001, N = 1078 wild-type [WT] vesicles, 1886 HTT-SD vesicles, and 1384 HTT-SA vesicles), retrograde velocities (ns: non-significant; N=1029 WT vesicles, 1564 HTT-SD vesicles, 2019 HTT-SA vesicles) and directional net flux (****p<0.0001; N=118 WT axons, 157 HTT-SD axons, 132 HTT-SA axons) of VAMP2-mCherry vesicles. Histograms represent means ± SEM of three independent experiments. Significance was determined using one-way ANOVA followed by Dunn's multiple comparison test. (**C**) Schematic of the three-compartment microfluidic device. Cortical neurons were infected with a lentivirus expressing VGLUT1 linked to a pH-sensitive variant of GFP (pHluorin); they were stimulated with 4AP-bicuculline at day in vitro (DIV) 11. The number of VGLUT-1 pHluorin exocytosis events within the synaptic chamber of the corticostriatal network, as compared to that of WT and to that of non-stimulated condition is shown here (*p<0.05; N=6712 events in WT, 4640 events in HTT-SD and 5176 events in HTT-SA neurons). The box-whisker plots show the median, the 25th and the 75th percentiles, the smallest and the largest values of three independent experiments using a total of N=WT 11, 10 HTT-SD, and 10 HTT-SA neurons seeded within microfluidic devices with at least three fields per device. Significance was determined using one-way ANOVA followed by Dunn's multiple comparison test.

The online version of this article includes the following video, source data, and figure supplement(s) for figure 1:

**Source data 1.** Data analyzed for anterograde velocity.

**Source data 2.** Data analyzed for retrograde velocity.

**Source data 3.** Data analyzed for net flux.

**Source data 4.** Data analyzed for VGLUT1 pHluorin exocytosis number of events.

**Figure supplement 1.** HTT phosphorylation at S421 increases synaptic vesicle precursor (SVP) anterograde axonal transport without affecting the quantity of SV released.

**Figure supplement 1—source data 1.** Data analyzed for number of anterograde vesicles.

**Figure supplement 1—source data 2.** Data analyzed for number of retrograde vesicles.

**Figure supplement 1—source data 3.** Data analyzed for the linear flow rate.

*Figure 1 continued on next page*

*Figure 1 continued*

**Figure supplement 1—source data 4.** Data analyzed for VGLUT1 pHluorin exocytosis amplitude of events.

**Figure 1—video 1.** Movie showing the glutamate release (VGLUT-pHluorin) in wild-type (WT) neurons after the stimulation.

https://elifesciences.org/articles/81011/figures#fig1video1

unphosphorylatable form of HTT is obtained by mutating the serine into an alanine (S421A) (*Thion et al., 2015*). We transduced cortical presynaptic neurons at DIV 0 (day in vitro 0) with a lentivirus encoding the major SNARE protein of SVs (v-SNARE), VAMP2 fused to mCherry protein (VAMP2-mCherry), a member of the vesicle-associated membrane protein (VAMP)/synaptobrevin family which labels SVPs (*Pennuto et al., 2003*). The circuit achieves functional maturity—as defined by the kinetics of neurite outgrowth, synapse formation, neuronal transport, and synchronous activity— by DIV 10–12 and establishes functional excitatory connections transmitting information from cortical to striatal neurons (*Moutaux et al., 2018*). We therefore performed all experiments in the microfluidic devices at this time point. We used high-resolution spinning confocal videomicroscopy to record VAMP2-mCherry particles in the distal part of cortical axons (*Figure 1A*) and generated kymographs to trace the movement of vesicles (*Figure 1A*, right panels).

Constitutive HTT phosphorylation (S421D) increased the anterograde velocity of VAMP2-positive vesicles (*Figure 1B*, left graph), the number of anterograde vesicles, and the linear flow rate (*Figure 1—figure supplement 1A*), leading to an increase in the net directional flux of VAMP2-mCherry vesicles traveling toward the presynapse (*Figure 1B*, right graph). There was no significant effect on VAMP2-mCherry vesicle velocities or net directional flux of HTT-SA mutation in axons (*Figure 1B*). HTT phosphorylation at S421 therefore influences transport of SVPs toward the presynapse.

We next investigated whether this increased presynaptic anterograde transport of SVPs affects the capacity of presynaptic neurons to release glutamate from SVs at the corticostriatal synapses. We transduced presynaptic cortical neurons with a lentivirus encoding the indicator of vesicle release and recycling VGLUT1-pHluorin, thanks to the fusion of a pH-sensitive GFP (pHluorin) to the vesicular glutamate transport VGLUT1 (*Fernandez-Alfonso and Ryan, 2008*). We then treated the presynaptic compartment with 4AP/bicuculline at DIV 10–12 to induce neuronal activity and measured the number of exocytic events per active synapse by recording fluorescence within the synaptic compartment (*Figure 1C*, *Figure 1—video 1*). The amplitude of VGLUT1 events was similar in WT, HTT-SA, and HTT-SD neurons (*Figure 1—figure supplement 1B*), but the frequency of release events at synapses was significantly greater only in HTT-SD neurons (*Figure 1C*). HTT phosphorylation thus promotes axonal transport of SVPs and increases the capacity of synapses to release glutamate.

## HTT constitutive phosphorylation at S421 impairs motor skill learning in mice

We had previously characterized HTT-SA, HTT-SD, and WT mice and found no differences between them in motor coordination (*Figure 2—figure supplement 1A*), forelimb strength (grip test), or anxious-depressive behavior (in the elevated plus maze test) (*Ehinger et al., 2020*). We therefore decided to reassess the three genotypes with a focus on the more subtle process of motor learning. To this purpose, we followed the mice daily as they developed skill on the rotarod (10 sessions per day for 8 consecutive days, for a total of 80 sessions) (*Figure 2A*). Running is largely hard-wired in mice, but WT mice do become more adept at staying on the rod with training, so this presented a suitable test for the improvement of a basic motor skill. Over the 8 days, the WT mice nearly doubled their latency to fall over the first 3 days/30 sessions of training (the learning phase) and then maintained that skill (the consolidation phase) (*Figure 2B*). The HTT-SD mice did not improve as much during the initial learning phase and plateaued at a lower level of skill. The HTT-SA mice learned much more gradually than WT but by the eighth day were virtually equivalent.

To better characterize this motor learning deficit, we examined the first and last days of training more closely. HTT-SD had greater difficulty adjusting to the rod initially than WT mice but improved by the end of the day (*Figure 2C*). On day 8 the plateau noted above was in evidence throughout the sessions (*Figure 2D*). The pattern with HTT-SA mice was less clear, at both 4 months (*Figure 2C and D*) and 18 months (*Figure 2—figure supplement 1B*).

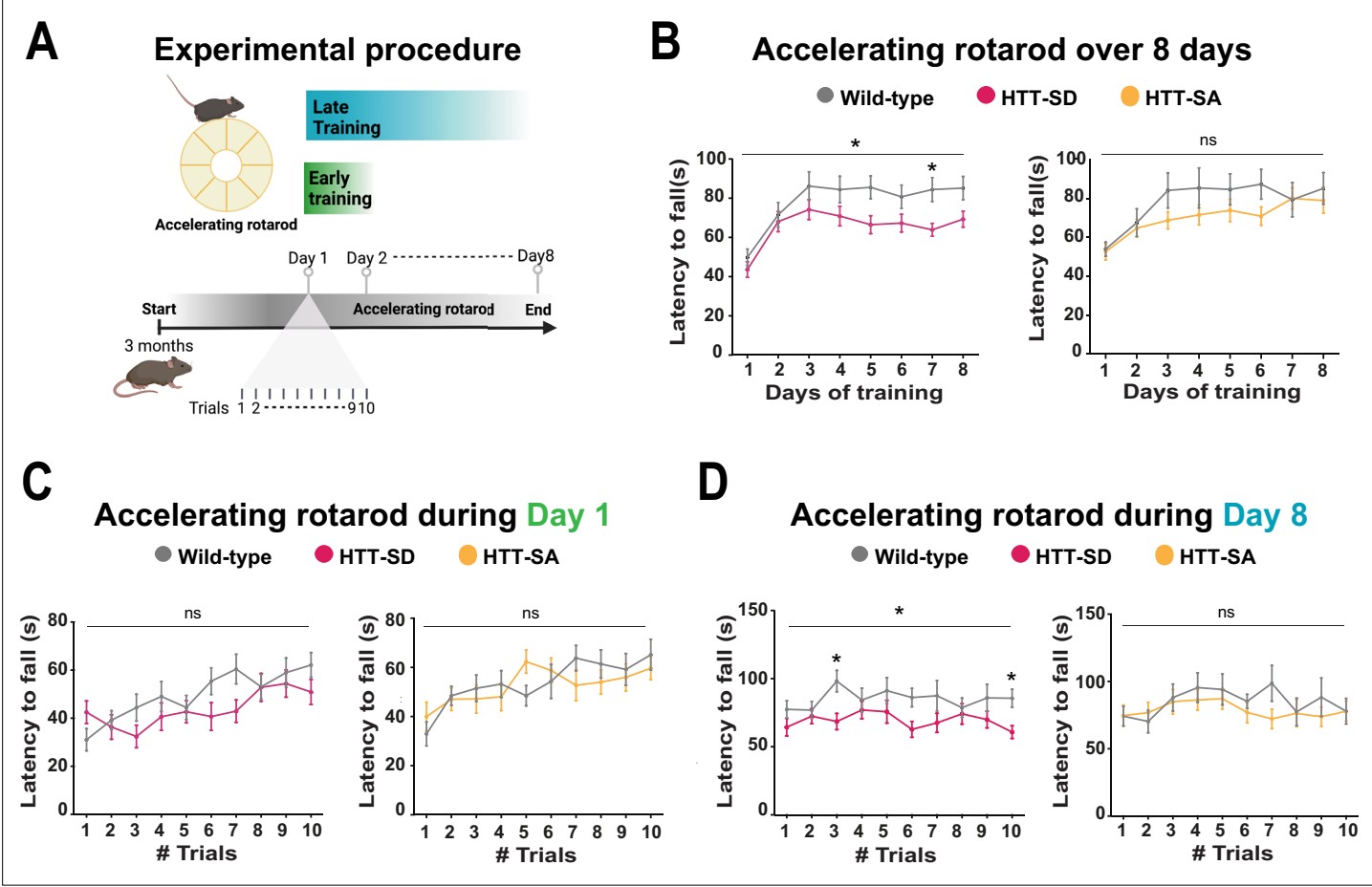

**Figure 2.** Constitutive phosphorylation of HTT at S421 impairs motor skill learning in mice. (**A**) Schematic of the accelerating rotarod protocol assessing motor skill learning over 8 days with 10 sessions per day. (**B**) Mean latency to fall each day for 8 days, for HTT-SD mice (*p<0.05: two-way ANOVA followed by Sidak's multiple comparisons test where p<0.01 at day 7) and HTT-SA mice (ns: non-significant: two-way ANOVA). (**C**) Mean time to fall off the rotarod per session, over 10 sessions, during the first or (**D**) the last day for HTT-SD (ns: non-significant: two-way ANOVA at day 1 and *p<0.05: two-way ANOVA at day 8 followed by Sidak's multiple comparison test where p<0.01 at trials 3 and 10) and HTT-SA (ns: non-significant: two-way ANOVA at days 1 and 8). We compared 3-month-old male mice: 20 wild-type (WT) with 20 HTT-SD littermates, and 13 WT with 18 HTT-SA littermates.

The online version of this article includes the following source data and figure supplement(s) for figure 2:

**Source data 1.** Data analyzed for accelerating rotarod over 8 days.

**Source data 2.** Data analyzed for accelerating rotarod during day 1.

**Source data 3.** Data analyzed for accelerating rotarod during day 8.

**Figure supplement 1.** HTT phosphorylation at S421 impairs motor skill learning without affecting motor performance.

**Figure supplement 1—source data 1.** Data analyzed for accelerating rotarod first trial at day 1.

**Figure supplement 1—source data 2.** Data analyzed for accelerating rotarod during day 1 at 18 months.

## HTT constitutive phosphorylation alters short-term plasticity

Motor skill learning relies on communication between the dorsal striatum and layer V pyramidal neurons in the motor cortex via the release of glutamate by the cortical afferences (*Graybiel and Grafton, 2015*; *Jin and Costa, 2015*; *Perrin and Venance, 2019*; *Yin and Knowlton, 2006*). We therefore performed whole-cell recordings of medium-sized spiny neurons (MSNs) from the dorso-lateral striatum (DLS) in acute corticostriatal brain slices from WT, HTT-SD, and HTT-SA adult mice (*Figure 3A*; see Materials and methods) and analyzed the spontaneous excitatory postsynaptic currents (sEPSCs). The HTT-SD mice did not differ from WT or HTT-SA mice in sEPSC amplitude or frequency (*Figure 3—figure supplement 1*). We next recorded EPSCs evoked by paired-pulse stimulations of layer V cortical neurons from the somatosensory S2 cortex and the corresponding

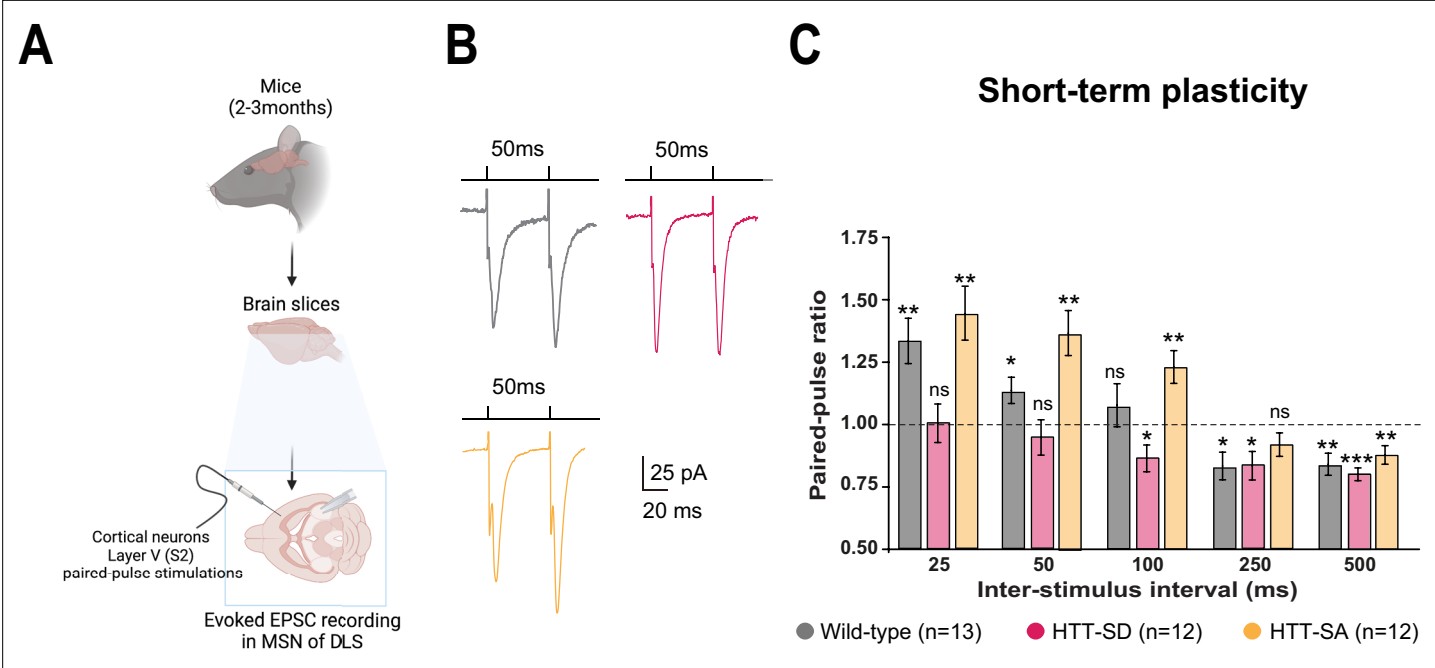

**Figure 3.** HTT phosphorylation at S421 increases short-term plasticity in the corticostriatal network ex vivo. (**A**) Schematic of medium-sized spiny neurons (MSNs) recording in the dorsolateral striatum (DLS) after paired-pulse stimulations in S2 cortex of mice at 2–3 months of age. (**B**) Representative traces of the paired-pulse ratio per interstimulus interval of electrophysiological response of MSNs in the DLS after stimulation in S2 in 2- to 3-month-old wild-type (WT) (gray), HTT-SD (pink), and HTT-SA (orange) mice (**C**). Quantification of (**B**). In contrast to WT and HTT-SA MSNs, HTT-SD MSN responses from 25 to 50 ms showed no facilitation (paired-pulse ratio~1) but only depression from 100 ms (*p<0.05, **p<0.001, and ***p<0.0001; ns means non-significant). Paired-pulse ratios were recorded from 13 WT, 12 HTT-SD, and 12 HTT-SA MSNs from at least N=3 mice.

The online version of this article includes the following source data and figure supplement(s) for figure 3:

**Source data 1.** Data analyzed for short-term plasticity.

**Figure supplement 1.** HTT phosphorylation does not regulate the spontaneous excitatory postsynaptic currents (sEPSCs) in the corticostriatal synapse.

**Figure supplement 1—source data 1.** Data analyzed for mEPSC recordings.

corticostriatal projection field in the dorsal striatum at various interstimulus intervals (ISIs: 25, 50, 100, 250, and 500 ms) to assess the probability of release at MSN corticostriatal synapses in WT, HTT-SD, and HTT-SA mice (*Figure 3*). Paired-pulse ratio (PPR) analysis revealed that in WT mice, corticostriatal short-term plasticity was facilitated for short ISIs (25 and 50 ms), with a lack of significant plasticity at 100 ms followed by a short-term depression for longer ISIs (250 and 500 ms), as previously described (*Goubard et al., 2011*). In HTT-SD mice, there was no short-term facilitation but only depression starting from 100 ms ISIs. HTT-SA mice showed a short-term plasticity similar to WT mice, except that the facilitation expression window widened up to 100 ms ISIs (*Figure 3*). Thus, while all genotypes showed similar short-term depression, they exhibited marked differences in facilitation.

The lower facilitation in HTT-SD MSNs indicates that constitutive phosphorylation of HTT increases the probability of glutamate release in pyramidal cells. These findings are in agreement with the greater number of exocytic events in HTT-SD neurons within microfluidic devices (*Figure 1C*).

## HTT phosphorylation increases the number and density of SVs at corticostriatal synapses

Since decreased facilitation could indicate changes in the number of SVs at the presynapse (*Park et al., 2012*; *Pulido and Marty, 2017*), we investigated the number of SVs at axon terminals within the corticostriatal network using electron microscopy. We focused on synapses formed between cortical neurons from the somatosensory cortex connecting with neurons from the DLS. According to the morphology of both the spines and the synapses, we counted the number of SVs in glutamatergic afferences within the DLS (*Figure 4A*). With chronic HTT phosphorylation (HTT-SD mice), the number of synapses did not change (*Figure 4Bi*) but there were a greater number of SVs at the axon terminals

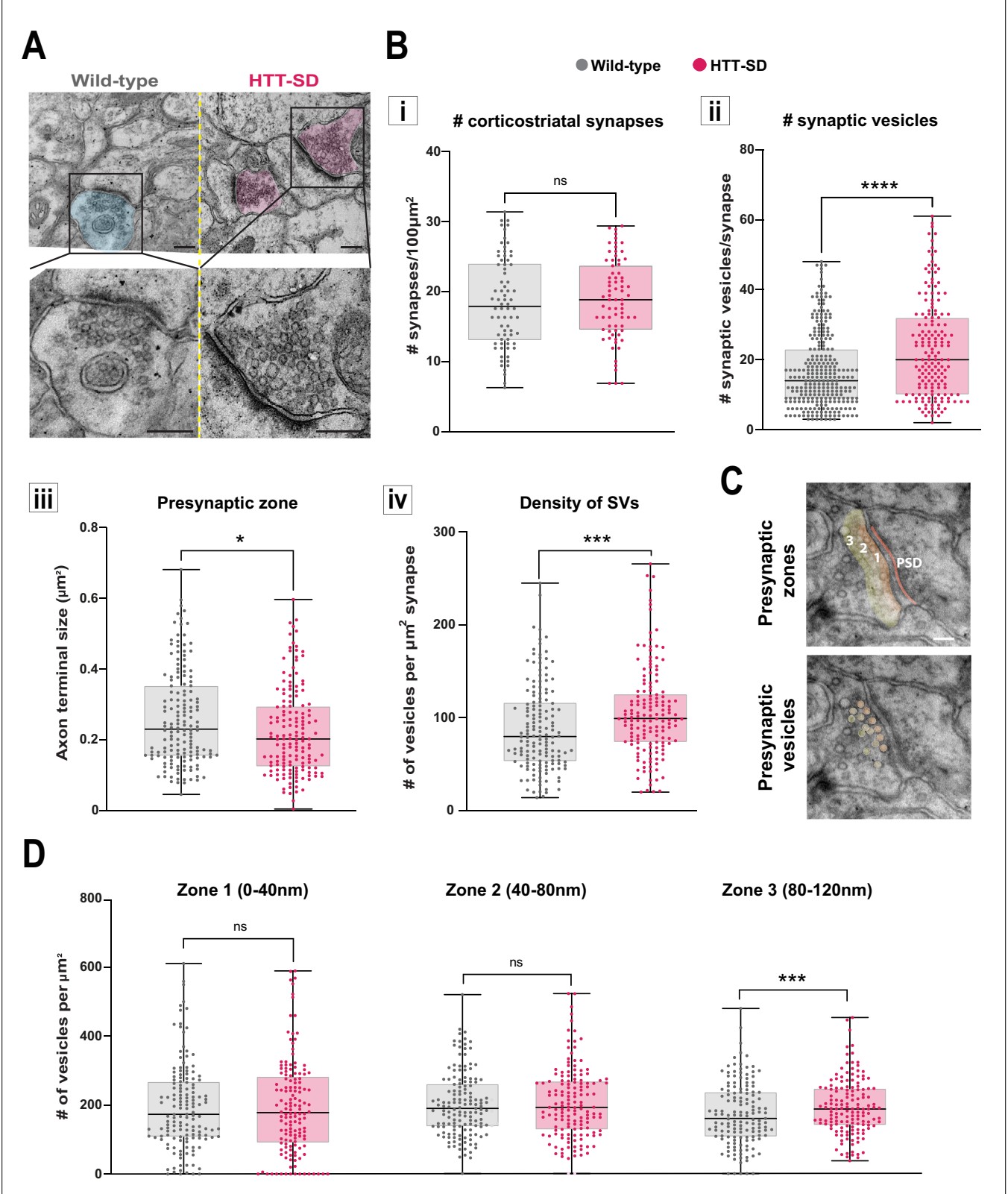

**Figure 4.** HTT phosphorylation increases the number of synaptic vesicles (SVs) distally to the presynaptic active zone. (**A**) Representative images of SVs at the corticostriatal synapse, obtained by electronic microscopy, in dorsolateral striatum (DLS) slices from three wild-type (WT) and HTT-SD mice 3-month-old male. Scale = 200 nm. (**B**) Quantification of (**i**) the number of synapses at the corticostriatal synapse per 100 µm² in DLS on n=74 WT and 74 HTT-SD striatal areas (ns: non-significant), (**ii**) the number of SVs per corticostriatal synapse from five WT and three HTT-SD mouse brains (N=279 WT

*Figure 4 continued on next page*

*Figure 4 continued*

and 171 HTT-SD axon terminals; ****p<0.0001), (iii) size of the cortical axon terminal in 158 WT and 156 HTT-SD corticostriatal synapses (*p<0.05), and (iv) the density of SVs within these axon terminals (number of vesicles per µm$^2$) in N=157 WT and 162 HTT-SD corticostriatal synapses (***p<0.001). (**C**) Representative images showing the 40-nm-wide zones in the axon terminal. Zone 1 is the closest to the synaptic cleft and contains the active zone. Zone 2 (40–80 nm) is adjacent to zone 1, and zone 3 (80–120 nm) is farthest from the active zone. Dark orange denotes the PSD within the striatal postsynaptic element. Scale = 100 nm. (**D**) The number of SVs per zone within the distal 120 nm of the axon terminal in at least 149±2 axon terminals (ns: non-significant, *p<0.05). The box-whisker plots show the median, the 25th and the 75th percentiles, the smallest and the largest value from at least three brains for each condition. Significances were determined using the Mann-Whitney test.

The online version of this article includes the following source data and figure supplement(s) for figure 4:

**Source data 1.** Data analyzed for the number of the corticostriatal synapses (i).

**Source data 2.** Data analyzed for the number of the synaptic vesicles (ii).

**Source data 3.** Data analyzed for the number of the axon terminal size in the presynaptic zone.

**Source data 4.** Data analyzed for the density of synaptic vesicles.

**Source data 5.** Data analyzed for the number of vesicles per zone.

**Figure supplement 1.** Characterization of the three presynaptic zones of HTT-SD corticostriatal axon terminals and analysis of HTT-SA corticostriatal synapses.

**Figure supplement 1—source data 1.** Data analyzed for the area of the presynaptic zones.

**Figure supplement 1—source data 2.** Data analyzed for the the number of the corticostriatal synapses (i).

**Figure supplement 1—source data 3.** Data analyzed for the number of the synaptic vesicles (ii).

than in WT mice (*Figure 4Bii*). The presynaptic size was smaller in the HTT-SD corticostriatal synapses (*Figure 4Biii*), leading to an increase in SV density in HTT-SD presynaptic terminals (*Figure 4Biv*). Although the total vesicle number may not be the major point of control for neurotransmission (*Fernandez-Alfonso and Ryan, 2008*; *Fredj and Burrone, 2009*; *Südhof, 2012*) and vesicle pools are not anatomically segregated (*Denker et al., 2009*; *Rizzoli, 2014*), it is worth noting that the most proximal zone is likely to contain the readily releasable pool (RRP) while the most distal zone is likely to be enriched in vesicles from the reserve pool (RP) (*Figure 4C*). We therefore divided the PSD into three 40-nm-wide zones so that we could measure vesicle density in each region. There was no difference between the genotypes in terms of the area of the zones (*Figure 4—figure supplement 1A*), but HTT-SD axon terminals exhibited a greater density of SVs in the most distal zone (zone 3) (*Figure 4D*). This indicates that constitutive HTT phosphorylation favors the anterograde transport of SVPs, leading to the accumulation of SVs in the distal presynapse in vivo. The fact that HTT-SA axon terminals had fewer vesicles lends further support to this explanation (*Figure 4—figure supplement 1B*).

Although we did not observe less anterograde transport or preferential retrograde transport of VAMP2-mCherry in our experimental conditions (*Figure 1B*), previous reports have noted a preferential retrograde trafficking of vesicles in HTT-SA mice (*Bruyère et al., 2020*; *Colin et al., 2008*; *Ehinger et al., 2020*). This suggests that in our in vitro experiments, WT HTT is phosphorylated at low levels. Since HTT-SD, but not HTT-SA, showed significant differences from WT in axonal transport, glutamate release, and motor skill learning, we focus on HTT-SD mice for the rest of this study.

## HTT recruits KIF1A to vesicles

The anterograde transport of SVP is driven predominantly by the kinesin-3 motor KIF1A (*Guedes-Dias and Holzbaur, 2019*). HTT and KIF1A interactomes suggested a possible interaction between the two proteins (*Shirasaki et al., 2012*; *Stucchi et al., 2018*), but this had not been tested. We observed KIF1A in the proteome of HTT-associated vesicles (*Figure 5A*; *Migazzi et al., 2021*). We found that KIF1A colocalizes with HTT immunopositive puncta in free-cultured cortical neurons at DIV 5 using a two-dimensional stimulated emission depletion (2D-STED) super-resolution microscope (*Figure 5B*, left panel). We confirmed this observation in HTT-SD neurons (*Figure 5B*, right panel). This suggests that phosphorylation of HTT could determine KIF1A recruitment on SVP.

We assessed this possibility by permeabilizing isolated axons within the distal part of microfluidic axonal compartments and, using Airyscan confocal high-resolution microscopy, we observed greater colocalization of HTT with KIF1A and VAMP2 in HTT-SD neuronal circuits than in WT (*Figure 5C* and *Figure 5—figure supplement 1A*). We then used proximity ligation assay (PLA) to confirm the in cellulo interaction between HTT and KIF1A. We observed a significantly greater PLA signal in HTT-SD

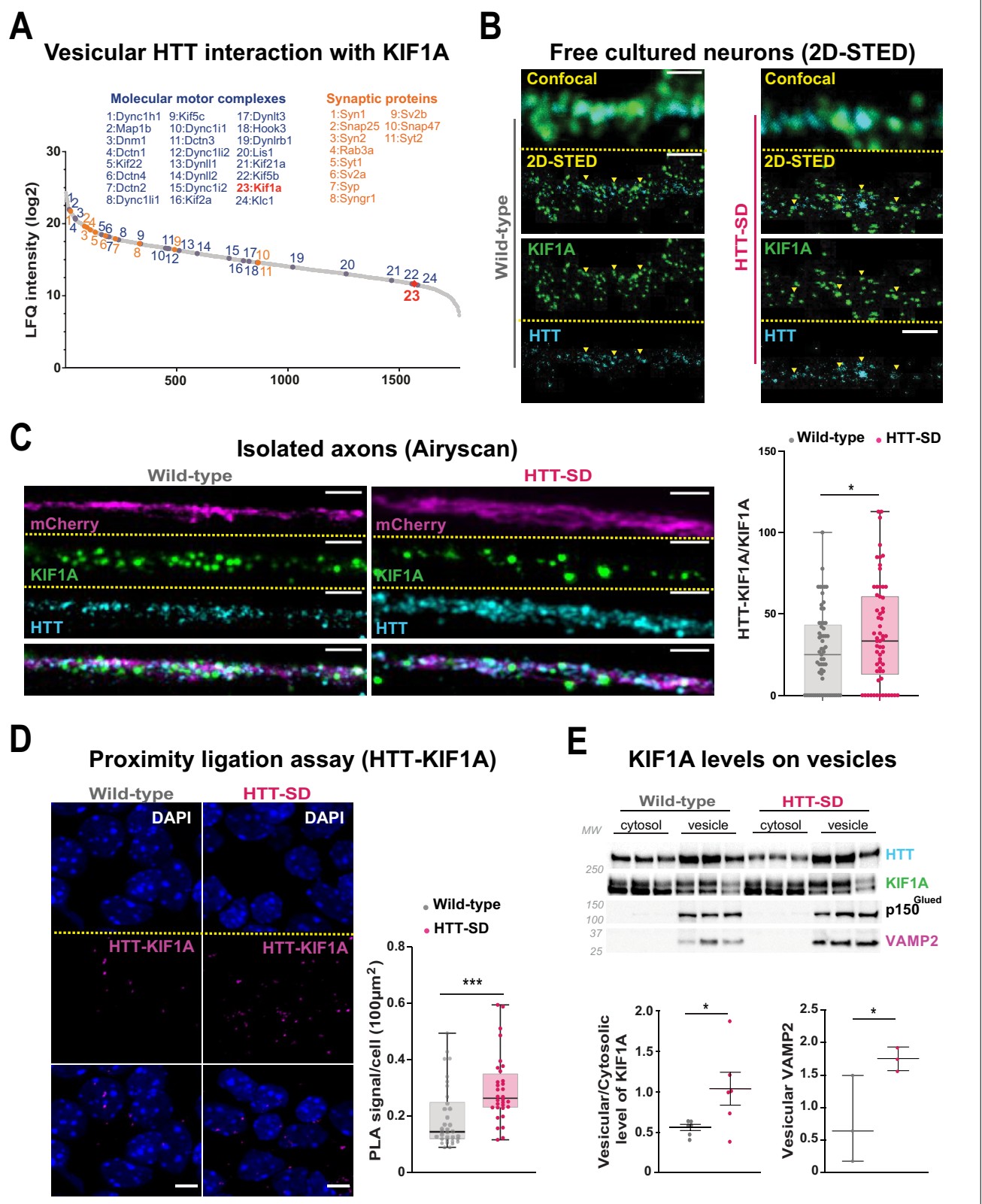

**Figure 5.** HTT phosphorylation recruits KIF1A on VAMP2-mCherry vesicles. (**A**) Mass spectrometry analysis of vesicles purified from mouse brains identifies KIF1A (red) among HTT-associated vesicular proteins. (**B**) Confocal and two-dimensional stimulated emission depletion (2D-STED) images of free-cultured neurons at day in vitro (DIV) 5 showing the colocalization of KIF1A and HTT. Scale bar: 1 μm. (**C**) Representative immunofluorescence labeling revealing HTT (cyan), KIF1A (green), and VAMP2-mCherry (magenta) within wild-type (WT) and HTT-SD cortical axons in the long channels of

*Figure 5 continued on next page*

*Figure 5 continued*

the microfluidic devices. The images were acquired in a specific region of interest and processed by an Airyscan detector (scale bar: 1 µm). Distribution analysis shows that HTT and KIF1A were more likely to colocalize on KIF1A⁺ vesicles in the HTT-SD condition. The graph represents means ± SEM of three independent experiments reproducing a corticostriatal network of WT or HTT-SD neurons in at least three microfluidic devices per experiment. Significance determined by the Mann-Whitney test (*p<0.05; N=61). (**D**) Proximity ligation assay (PLA) in WT or HTT-SD neurons, nuclei stained with DAPI. Representative images are from three independent experiments. Scale bar: 10 µm. Significance was determined by the Mann-Whitney test (***p<0.0001; N=32–34). (**E**) Western blot analysis for HTT, KIF1A (both bands), p150^Glued, and tubulin from vesicular fractions from six WT and six HTT-SD brains. Significance was determined using the Mann-Whitney test (*p<0.05).

The online version of this article includes the following source data and figure supplement(s) for figure 5:

**Source data 1.** Data analyzed for HTT-KIF1A colocalization in axons.

**Source data 2.** Data analyzed for the proximity ligation assay performed between HTT and KIF1A.

**Source data 3.** Data analyzed for the protein content of KIF1A and VAMP2 levels in vesicular fractions.

**Source data 4.** Western blot scans for the data presented in *Figure 5E* (KIF1A and VAMP2 levels in brain vesicular fractions).

**Figure supplement 1.** HTT phosphorylation and subcellular localization and interaction of HTT and KIF1A with VAMP2.

**Figure supplement 1—source data 1.** Data analyzed for HTT-VAMP2 colocalization in axons.

**Figure supplement 1—source data 2.** Data analyzed for KIF1A-VAMP2 colocalization in axons.

**Figure supplement 1—source data 3.** Data analyzed for the proximity ligation assay performed between VAMP2/HTT.

**Figure supplement 1—source data 4.** Data analyzed for the proximity ligation assay performed between VAMP2/KIF1A.

**Figure supplement 1—source data 5.** Data analyzed for the protein content of KIF1A and VAMP2 levels in total fractions.

**Figure supplement 1—source data 6.** Western blot scans for the data presented in *Figure 5—figure supplement 1D* (KIF1A levels in whole brain lysates).

neurons (*Figure 5D*), despite no change in PLA levels of VAMP2 with HTT and KIF1A (*Figure 5—figure supplement 1B and C*). Finally, we prepared vesicular-enriched fractions from WT and HTT-SD mouse brains and immunoblotted them for KIF1A. HTT-SD showed a greater vesicular/cytosolic ratio for KIF1A (*Figure 5E*), while p150 remained constant and the total brain levels of KIF1A did not differ between the genotypes (*Figure 5—figure supplement 1D*). KIF1A and HTT thus colocalize on VAMP2-positive vesicles, and S421 phosphorylation augments KIF1A interaction with HTT.

## HTT-SD-mediated SVP transport depends on KIF1A

We next asked whether the phospho-HTT-mediated increase in SVP anterograde transport depends on KIF1A by using a validated sh-Kif1a (*Kevenaar et al., 2016*). Lentiviral expression of sh-Kif1a in cortical neurons reduced KIF1A expression by ~83% (*Figure 6—figure supplement 1A*). We then treated corticostriatal projecting neurons plated in microfluidic devices with lentiviruses expressing either sh-scramble-GFP (sh-Scr) or sh-Kif1a-GFP. We recorded axonal transport of VAMP2-mCherry vesicles at DIV 12 and generated kymographs as before (*Figure 6A*). We found that silencing KIF1A in WT cortical neurons decreased VAMP2 anterograde vesicle velocity, the number of anterograde vesicles (*Figure 6B* and *Figure 6—figure supplement 1C*), the linear flow (*Figure 6—figure supplement 1C*), and the net directional flux of VAMP2 vesicles toward the axon terminals (*Figure 6B*).

We next measured VAMP2 transport in HTT-SD neurons and found greater anterograde velocity, number of anterograde vesicles, and positive net directional flux (and/or linear flow) than in WT neurons (*Figure 6B* and *Figure 6—figure supplement 1C*), confirming our previous results (*Figure 1B*). Silencing KIF1A in HTT-SD reduced the anterograde velocity of VAMP2 vesicles close to values observed in WT neurons (*Figure 6B*). KIF1A silencing also reduced the number of antero-grade vesicles (*Figure 6—figure supplement 1B*), linear flow, and net directional flux in HTT-SD to values found in WT (*Figure 6B* and *Figure 6—figure supplement 1B*). The velocity and number of retrograde-moving VAMP2 vesicles was also lower in HTT-SD neurons (*Figure 6B*). This attenuation of retrograde transport might be linked to KIF1A's reported role as a dynein activator (*Chen et al., 2019*).

We considered the possibility that the observed increase in SV release might be due in part to a synergistic action of BDNF at synapses, both because of the prominent role of HTT and its phos-phorylation in regulating BDNF transport (*Colin et al., 2008*; *Ehinger et al., 2020*; *Gauthier et al., 2004*) and because synaptic BDNF levels regulate synaptic plasticity and SV release (*Gangarossa*

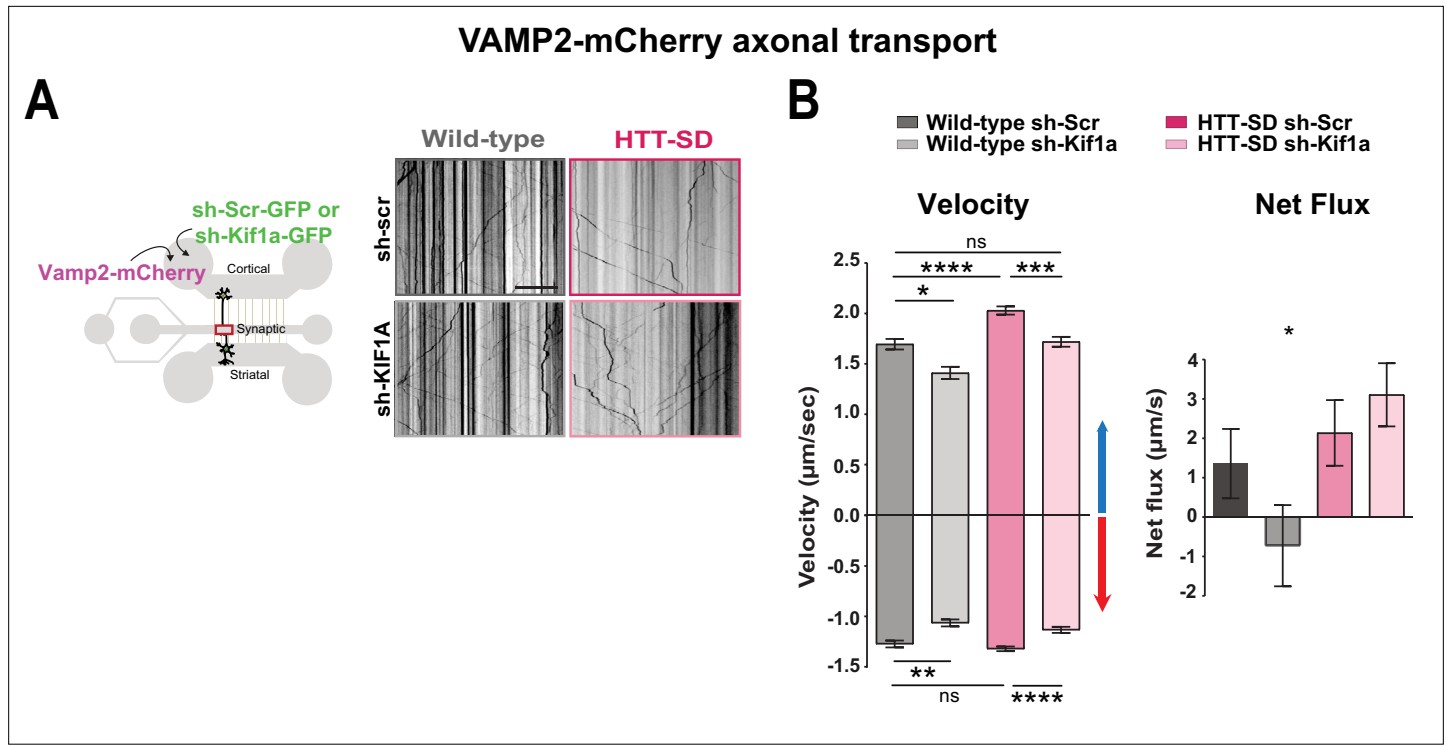

**Figure 6.** HTT-dependent axonal transport of synaptic vesicle precursors (SVPs) is mediated by KIF1A. (**A**) Diagram indicating lentiviral transduction of VAMP2-mCherry and sh-scramble (sh-Scr-GFP) or sh-Kif1a (sh-Kif1a-GFP) lentiviruses at day in vitro (DIV) 8 in the microfluidic device. On the right, representative kymographs of VAMP2-mCherry vesicle transport in axons for each condition. Scale bar = 25 μm. (**B**) Segmental anterograde and retrograde velocities (anterograde: *p<0.05, ***p<0.001, ****p<0.0001; N=548 vesicles wild-type [WT] sh-Scr, 318 vesicles WT sh-Kif1a, 1129 vesicles HTT-SD sh-Scr, 628 vesicles HTT-SD sh-Kif1a) (retrograde: *p<0.05, **p<0.01, ****p<0.0001; N=583 vesicles WT sh-Scr, 396 vesicles WT sh-Kif1a, 1282 vesicles HTT-SD sh-Scr, 620 vesicles HTT-SD sh-Kif1a) and directional net flux (*p<0.01; N=79 axons WT sh-Scr, 59 axons WT sh-KIFA,112 axons HTT-SD sh-Scr, 89 axons HTT-SD sh-Kif1a; one-way ANOVA test) of VAMP2-mCherry vesicles in WT and HTT-SD neurons transduced with sh-Scr or sh-Kif1a lentiviruses. Histograms represent means ± SEM of three independent experiments. Significance was determined using a one-way ANOVA followed by Dunn's multiple comparison test.

The online version of this article includes the following source data and figure supplement(s) for figure 6:

**Source data 1.** Data analyzed for anterograde velocity.

**Source data 2.** Data analyzed for retrograde velocity.

**Source data 3.** Data analyzed for net flux.

**Figure supplement 1.** KIF1A levels in HTT-SD neurons regulate VAMP2 axonal transport.

**Figure supplement 1—source data 1.** Western blot scans for the data presented in *Figure 6—figure supplement 1A* (KIF1A levels in cortical neurons).

**Figure supplement 1—source data 2.** Data analyzed for number of anterograde vesicles.

**Figure supplement 1—source data 3.** Data analyzed for number of retrograde vesicles.

**Figure supplement 1—source data 4.** Data analyzed for the linear flow rate.

**Figure supplement 2.** KIF1A silencing doesn't affect BDNF-mCherry transport.

**Figure supplement 2—source data 1.** Data analyzed for for anterograde velocity.

**Figure supplement 2—source data 2.** Data analyzed for for retrograde velocity.

**Figure supplement 2—source data 3.** Data analyzed for the linear flow rate.

**Figure supplement 2—source data 4.** Data analyzed for the net flux.

*et al., 2020*; *Park and Poo, 2013*; *Park et al., 2014*; *Tyler et al., 2006*; *Walz et al., 2006*). Furthermore, dense-core vesicles, including those containing BDNF, can be transported by kinesin-3 (*Hung and Coleman, 2016*; *Lim et al., 2017*; *Lo et al., 2011*; *Stucchi et al., 2018*). We therefore silenced KIF1A in cortical axons and measured BDNF-mCherry axonal transport in the distal part of axons at DIV 12 (*Figure 6—figure supplement 2A*). Chronic HTT phosphorylation increased the anterograde

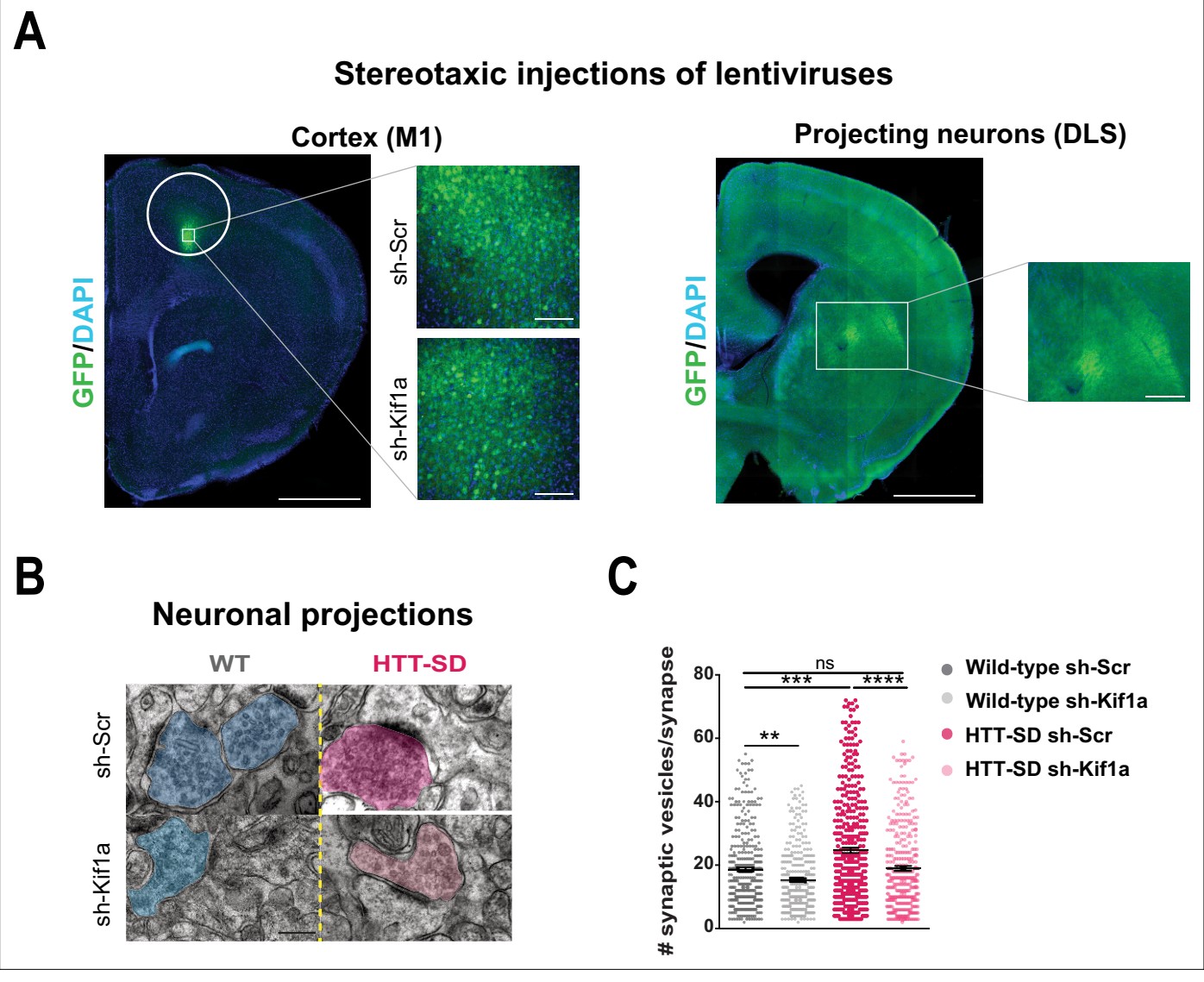

**Figure 7.** In vivo KIF1A silencing in mice restores the synaptic vesicle (SV) synaptic pool. (**A**) Immunolabeling of GFP within the injection site on a slice located at 1.5 mm before the bregma (left). Scale = 1 cm (insets, 100 μm). Immunolabeling of GFP within the projection site on a slice located at –0.3 mm after the bregma (right). Scale = 1 cm (inset, 250 μm). Nuclei are labeled with DAPI. (**B**) Representative images from electron microscopy of corticostriatal synapses and (**C**) quantification of the number of SVs at the corticostriatal synapse of three wild-type (WT) male mice injected with either sh-Scr or sh-Kif1a and three HTT-SD mice injected with sh-Scr or sh-Kif1a (**p<0.01, ***p<0.001, ****p<0.0001; N=360 WT sh-Scr synapses, 324 WT sh-Kif1a synapses, 417 HTT-SD sh-Scr synapses, 337 HTT-SD sh-Kif1a synapses). Scale = 200 nm. Histograms represent means ± SEM. Significance was determined using one-way ANOVA followed by Dunn's multiple comparison test.

The online version of this article includes the following source data for figure 7:

**Source data 1.** Data analyzed for the number of the synaptic vesicles.

transport, the linear flow, and the net directional flux of BDNF-containing vesicles, in agreement with previous studies (*Colin et al., 2008*; *Ehinger et al., 2020*; *Figure 6—figure supplement 2B*). Silencing KIF1A did not affect BDNF dynamics either in WT or in HTT-SD neurons. This indicates that the phospho-HTT-dependent increase in SVP anterograde transport, in contrast to BDNF, is mediated by KIF1A.

## HTT-KIF1A-mediated transport regulates the number of SVs at synapses

We next investigated whether SVP anterograde transport via the HTT-KIF1A complex regulates the number of vesicles at synapses. We injected lentiviruses encoding either sh-scramble-GFP or sh-Kif1a-GFP into layer V of the HTT-SD motor cortex (*Figure 7A*), whose neurons project mainly to the DLS (*Hunnicutt et al., 2016*). We then counted the number of SVs at corticostriatal synapses from sections prepared from WT and HTT-SD brains injected with lentiviral sh-Scr or sh-Kif1a. WT sh-Kif1a presynapses showed significantly fewer SVs than WT sh-Scr presynapses (*Figure 7C*). As previously shown (*Figure 4B*), there were significantly more SVs at presynapses in HTT-SD, but this number reverted to WT levels in HTT-SD brains treated with sh-Kif1a (*Figure 7C*). These data demonstrate that decreasing the phospho-HTT-mediated anterograde transport of SVPs by reducing KIF1A levels in corticostriatal-projecting neurons reduces the number of SVs at presynapses. This in turn indicates a close relationship between axonal transport and synaptic SV content.

## HTT-KIF1A-mediated axonal transport of SVPs in corticostriatal-projecting neurons regulates motor skill learning

To determine whether the modification in anterograde transport via the HTT-KIF1A complex within corticostriatal-projecting neurons is responsible for the defect in motor skill learning we observed in HTT-SD mice, we injected lentiviral vectors encoding sh-Scr-GFP and sh-Kif1a-GFP into 3-month-old WT and HTT-SD mice. Three weeks later, we subjected the mice to the same rotarod protocol as before (*Figure 8A*). HTT-SD mice did not show much improvement over 8 days (*Figure 8B*), as with the non-injected mice (*Figure 2B*). Silencing KIF1A improved the performance of the HTT-SD mice over the first 6 days, but then the mice seemed to lose ground (*Figure 8B*). Indeed, the improvement in motor learning of the HTT-SD mice via sh-Kif1a silencing was significant on the first day of training, but the effect did not last until the eighth day (*Figure 8C*). This could be related to the duration of gene silencing in nondividing cells that is about 3 weeks while our experimental procedure extent up to 4 weeks (*Bartlett and Davis, 2006*). Nonetheless, these findings indicate that HTT-KIF1A-mediated axonal transport of SVPs in the corticostriatal projecting neurons, a process modulated by phosphorylation, influences the number of SVs at synapses, the probability of release, and the efficacy of motor skill learning.

## Discussion

We show here that axonal transport of SVPs influences synaptic function. Specifically, HTT's phosphorylation status fine-tunes SVP transport efficiency through its recruitment of KIF1A. Genetically blocking dephosphorylation at S421 impaired motor learning, and abolishing KIF1A activity in the context of constitutive phosphorylation only partially restored motor performance on the rotarod. Such a genetic approach is rather blunt compared to the sensitivity of (de)phosphorylation in responding to cellular signals, yet it enabled us to answer the question that motivated the study and show that axonal transport does influence synaptic homeostasis, with consequences for circuit function and behavior.

### Huntingtin and the regulation of SVP axonal transport

This work places HTT among the proteins that participate in SVP transport (*Guedes-Dias and Holzbaur, 2019*) and closes a loop opened by the discovery that DENN/MADD, a Rab3-GEP that binds to KIF1A (and KIF1Bß), regulates SVP binding to microtubules according to Rab3's nucleotide state (*Niwa et al., 2008*). Rab3 is part of the HTT interactome (*Shirasaki et al., 2012*) and enriched in SVs (*Takamori et al., 2006*); previous work in *Drosophila* larval axons showed that reducing HTT levels decreases the transport of Rab3-positive vesicles (*White et al., 2015*). In the context of Huntington disease (HD), which is caused by polyglutamine expansions in HTT, both Rab3 levels and the conversion from GTP to GDP state are dysregulated, which is consistent with a role for HTT in the transport of SVPs.

Studies in HD models have revealed alterations in HTT phosphorylation at S421 that could result from defects in Akt, the S421 kinase, or dysregulation of the phosphatases calcineurin (PP2B) and PP2A (*Humbert et al., 2002*; *Metzler et al., 2010*; *Pardo et al., 2006*; *Warby et al., 2005*). Whether SVP trafficking is also altered and whether restorating SVP transport through HTT phosphorylation

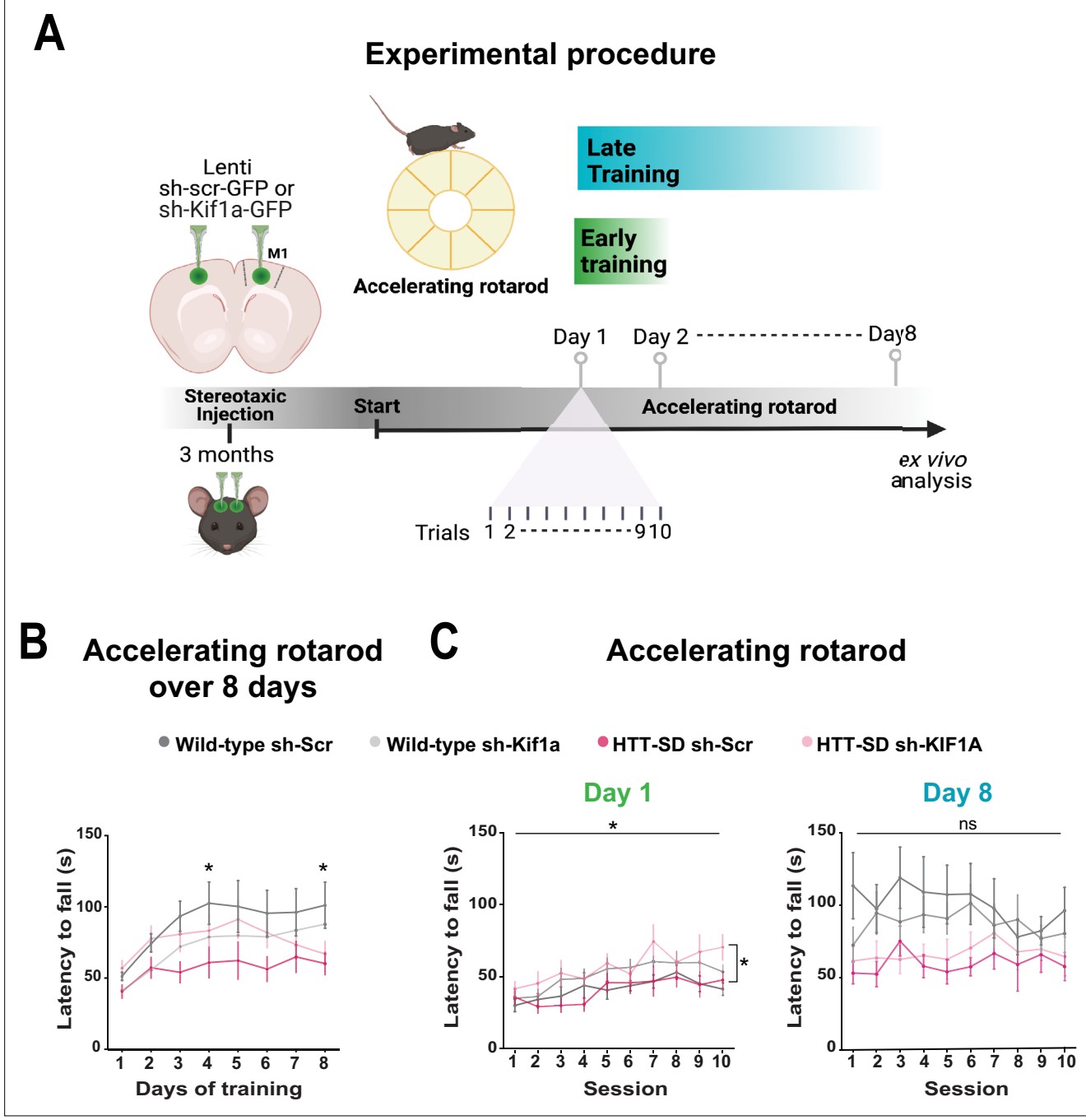

**Figure 8.** Motor skill learning defects of S421D mice are rescued by KIF1A silencing in vivo. (**A**) Schematic of the experimental procedure consisting in bilateral stereotaxic injections in the mouse brain followed 3 weeks later by the accelerating rotarod protocol over 8 days. (**B**) Mean time to fall off the rotarod per day over 8 days. Two-way ANOVA comparing the four conditions showed significant differences between genotypes and silencing conditions (****$p<0.0001$). Holm-Sidak's post hoc analysis revealed significant differences between wild-type (WT) sh-Scr and HTT-SD sh-Scr mice at day 4 and day 8 (*$p<0.05$). (**C**) Mean time to fall off the rotarod the first day (*$p<0.01$) (left) and the last day (ns: non-significant, right), per sessions. Holm-Sidak's post hoc analysis revealed significant differences between HTT-SD sh-Scr and HTT-SD sh-Kif1a mice during the first day (*$p<0.05$). At least three cohorts containing 12 WT sh-Scr, 11 WT sh-Kif1a, 10 HTT-SD sh-Scr, and 12 HTT-SD sh-Kif1a 3-month-old littermate male mice were analyzed.

The online version of this article includes the following source data for figure 8:

**Source data 1.** Data analyzed for accelerating rotarod over 8 days.

**Source data 2.** Data analyzed for accelerating rotarod during day 1.

**Source data 3.** Data analyzed for accelerating rotarod during day 8.

could mitigate HD pathology are questions that remain to be answered. That such a study would be worthwhile is suggested by the fact that promoting HTT phosphorylation is neuroprotective, as it restores the transport and release of BDNF (*Humbert et al., 2002*; *Kratter et al., 2016*; *Pardo et al., 2006*; *Pineda et al., 2009*; *Warby et al., 2009*; *Zala et al., 2008*). It is interesting to note in this context that release of glutamate at the corticostriatal synapse is altered in HD brains-on-chips (*Virlogeux et al., 2018*) and in vivo (*Fernández-García et al., 2020*; *Smith-Dijak et al., 2019*) and that stimulating glutamatergic corticostriatal connections in HD reverses motor symptoms in HD mice (*Fernández-García et al., 2020*), suggesting that reestablishment of glutamate synaptic release capacities could have therapeutic potential.

## Fine-tuning SVP transport regulates synapse homeostasis and proper neurotransmission

Several studies have linked a reduction in axonal anterograde transport of SVPs to a decline in synaptic function. Indeed, genetically impairing KIF1A reduces the number of SVs at nerve terminals and causes postnatal lethality (*Yonekawa et al., 1998*). KIF1A loss-of-function variants, most of them located within the conserved motor domain, reduce SVP transport and are associated with four diseases: autosomal recessive hereditary sensory neuropathy IIC, autosomal dominant mental retardation (ADMR) type 9, autosomal recessive spastic paraplegia type 30, and autosomal dominant hereditary spastic paraplegia (HSP) (*Pennings et al., 2020*). That all these disorders involve lower extremity spasticity and weakness reflects the challenge of transporting vesicles down the extremely long axons of the peripheral nervous system, but the cognitive deficits in ADMR type 9 show that disruptions in axonal transport clearly disturb synaptic transmission and synaptic strength in the central nervous system as well, with obvious consequences for learning and memory (*Guedes-Dias et al., 2019*; *Zhang et al., 2017*).

We found that increasing SVP axonal transport via phospho-HTT-mediated KIF1A activation is equally problematic: too many SVs in the synaptic pool are also detrimental to synaptic function and motor skill learning. This is consistent with the fact that the KIF1A gain-of-function mutation V8M, which causes another type of HSP, leads to greater anterograde transport of vesicles and abnormal accumulation of vesicles at synapses (*Chiba et al., 2019*; *Gabrych et al., 2019*). Further evidence for the sensitivity of synapses to both too little and too much SVPs comes from point mutations in the autoinhibitory domain of *unc-104* that cause hyperactive axonal transport and abnormal accumulation of SVs (*Cong et al., 2021*). Similarly, depletion of kinesin-binding protein, KBP, which inhibits KIF1A activity, leads to the abnormal accumulation of both KIF1A and vesicles at neurite terminals (*Kevenaar et al., 2016*), and nonsense mutations of KBP cause Shprintzen-Goldberg syndrome (also called marfanoid-craniosynostosis syndrome), which is characterized by intellectual disability, skeletal abnormalities, and axonal neuropathy (*Dafsari et al., 2015*; *Valence et al., 2013*). While we show here that S421 phosphorylation promotes anterograde transport and abnormal accumulation of SVs at the synapse, the lack of phosphorylation seemed to have only a limited effect on the transport and release of SVs in vitro. This is consistent with previous studies analyzing the transport of other cargoes, which revealed more obvious differences for anterograde than for retrograde transport (*Bruyère et al., 2020*; *Ehinger et al., 2020*). This could be due to the culture conditions in microfluidic devices, in which the level of phosphorylation of endogenous WT HTT is rather low. We nonetheless observed slight alterations in HTT-SA mice, including fewer SVs at the presynapse and a more gradual learning curve. Although less marked, these changes are the opposite of the alterations observed when HTT is consistently phosphorylated. Together, these studies indicate that SVP transport is normally fine-tuned to ensure the proper quantity of SVs at the synapse and effective synaptic function.

## Axonal SVP transport, the SV pool, and SV release probability

Within the presynapse, SVs are organized into different pools—the RRP, the recycling pool, and the RP—according to their composition, age, and distance from the active zone (*Crawford and Kavalali, 2015*; *Kaeser and Regehr, 2017*; *Rizzoli, 2014*; *Truckenbrodt et al., 2018*). The RRP contains the SVs ready to be released upon neural activity and is thought to be refilled by SVs from the RP by a slow process (hundreds of milliseconds) (*Pulido and Marty, 2017*; *Rizzoli, 2014*). We observed an increased number of vesicles in the pool most distal from the membrane. This is consistent with the fact that vesicles are most likely to accumulate where the SVPs detach from the KIF1A motor, away

from the membrane and the active zone (*Bodaleo and Gonzalez-Billault, 2016*). In addition, newly produced vesicles arriving at the presynaptic zone are preferentially released (*Truckenbrodt et al., 2018*).

Although the locations of SV populations relative to the membrane do not reliably define vesicular pool identities (*Denker et al., 2009*; *Fernandez-Alfonso and Ryan, 2008*; *Fredj and Burrone, 2009*), we propose that this population distal from the membrane influences the replenishment of the pools nearer the membrane, that is, the RRP (*Kidokoro et al., 2004*) and/or the recycling pool (*Ratnayaka et al., 2012*). All the vesicle pools are dynamic and interdependent (*Chamberland and Tóth, 2016*; *Chanaday et al., 2019*), and previous work in *Drosophila* has shown that impaired recruitment of SVs from the RP to the RRP causes memory deficits and limited short-term synaptic plasticity (*Kidokoro et al., 2004*). Modulation of the RP itself, in size or dynamicity, has also been shown to affect plasticity and memory (*Corradi et al., 2008*; *De Rossi et al., 2020*; *Gitler et al., 2008*; *Skorobogatko et al., 2014*).

## DLS and motor skill learning

We found that lowering KIF1A levels within M1 neurons that project mainly to the DLS rescued motor skill learning, a process that is usually attributed to the dorsomedial striatum (DMS) (*Costa et al., 2004*; *Yin et al., 2009*). Recent studies have suggested that both DMS and DLS are engaged in learning (*Bergstrom et al., 2018*; *Gremel and Costa, 2013*; *Kimchi et al., 2009*; *Kupferschmidt et al., 2017*; *Perez et al., 2022*; *Perrin and Venance, 2019*; *Stalnaker et al., 2010*; *Thorn et al., 2010*), so targeting the DLS might be sufficient to rescue the development of motor skill during the first days of rotarod training. This would not, however, explain the lack of rescue upon KIF1A silencing in HTT-SD mice during the consolidation phase, when WT mice show that they maintain the skill level that peaked at day 3. Rather, this could be related to the observation that during consolidation, the number of neurons that are activated upon stimulation of the DLS falls as the circuit streamlines its connections (*Badreddine et al., 2022*; *Cao et al., 2015*; *Picard et al., 2013*). In other words, as the motor skill is mastered, fewer neurons are needed to code the activity. Thus, it is possible that this small number of DLS connections responsible for consolidation of a learned skill would not have been targeted by the sh-Kif1a virus since it is unlikely that its expression would cover the entire DLS.

Altogether, our results highlight the importance of axonal SVP transport for synaptic transmission and identify a role for the HTT-KIF1A pathway as a regulator of SVP transport, synapse function, and motor skill learning within corticostriatal projecting neurons.

# Materials and methods

## Contact for reagent and resource sharing

Further information and requests for resources and reagents should be directed to and will be fulfilled by the Lead Contact, Frédéric Saudou (frederic.saudou@inserm.fr).

## Experimental model and subject details

### Mice

*Htt*^S421A/S421A and *Htt*^S421D/S421D mice (referred to as HTT-SA and HTT-SD mice, respectively) have been previously described (*Thion et al., 2015*). They were generated by the Mouse Clinical Institute (Strasbourg, France). Briefly, these C57BL/6J mice were knocked in with a point mutation replacing the serine 421 by an alanine or an aspartic acid, respectively. All mice were maintained with access to food and water ad libitum and kept at a constant temperature (19–22°C) and humidity (40–50%) on a 12:12 hr light/dark cycle. All experimental procedures were performed in an authorized establishment (Grenoble Institut Neurosciences, INSERM U1216, license #B3851610008) in strict accordance with the directive of the European Community (63/2010/EU). The project was approved by the French Ethical Committee (Authorization number: APAFIS#18126-2018103018299125 v2) for care and use of laboratory animals and performed under the supervision of authorized investigators. For behavior studies, only males were used at 3–4 months of age. Behavioral studies compared littermates, homozygous (*Htt*^S421DorA/S421DorA), or WT (*Htt*^+/+) mice. Those mice were then used for electrophysiological and biochemical studies or processed to be imaged by electron microscopy. The number of animals was limited to the minimum number necessary per group in order to have at least an 80% chance of

detecting a significant difference (power 1-β) and a risk of error α of 5%. This number was determined using a statistical test for estimating the optimal sample size using the variances determined in a preliminary study.

For biochemistry and neuronal culture (E15.5), the sex distinction of homozygous or WT mice was not made. Specific ages used for each experiment are indicated in the figure legends. C57BL/6J mice, purchased from Charles River Laboratory, were used for backcrosses to maintain the colony and to obtain WT E15.5 pups.

## Primary neuron culture and transduction

Primary cortical and striatal neurons were dissected from E15.5 WT (C57Bl/6J) or HTT-SA or HTT-SD mouse embryos as previously described (*Liot et al., 2013*). They underwent a chemical dissociation with papain cysteine solution, DNase (1/100), and FBS (1/10) and were finally mechanically dissociated. They were re-suspended in a growing medium containing a Neurobasal medium, 2% B27, 1% penicillin/streptomycin, and 2 mM Glutamax ($5 \times 10^6$ cells in 120 µl). Cortical neurons were plated in the presynaptic chamber coated with poly-D-lysine (0.1 mg/ml) and striatal neurons were plated in the postsynaptic chamber coated with poly-D-lysin and laminin (10 µg/ml) with a final density of ~7000 cells/mm². A growing medium was added to the synaptic chamber to equilibrate the flux. Neurons were left in the incubator for 2 hr and then all compartments were gently filled with growing medium. Neurons were cultured at 37°C in a 5% $CO_2$ incubator for 10–12 days.

Between DIV 0 and DIV 4, cortical neurons within the presynaptic compartment were transduced with lentiviruses expressing VAMP2-mCherry, VGLUT1-pHluorin, or BDNF-mCherry. Neurons were washed out the next day. At DIV 8, cortical neurons were transduced with sh-Kif1a or sh-Scr lentiviruses.

## Accelerating rotarod

Motor skill learning was assessed using an accelerating rotarod (LE8305, BIOSEB). The tests were performed during the beginning of the light phase on male littermates housed in the same cages. Cages were transported to the experimental room at least 30 min before the tests to allow habituation of the mice to the room kept at a constant temperature (19–22°C) and humidity (40–50%). The day preceding the test, the mice were acclimated to the rod by one session (1 min at 4 rpm). Then, the accelerating rotarod assay was performed over 8 consecutive days with 10 sessions per day per mouse, increasing the speed from 4 to 40 rpm over 5 min. Each trial was separated by at least a 15 min resting period. The latency and the speed to fall off from the rotarod were recorded.

## Stereotaxic injections

Three-month-old HTT-SD and WT male mice were anesthetized by inhalation of isoflurane associated with a mix of oxygen and room air (3–5% of isoflurane for induction and 1–2% in the mask). The mouse head was then shaved and placed within the stereotaxic frame. The skin was incised, and the skull was bilaterally drilled. The capillary was inserted slowly. We injected the brains bilaterally at the following coordinates: anterior-posterior axis, AP, +1.54 mm; medial-lateral axis, ML, +1.6 mm or –1.6 mm; and dorsal-ventral axis, DV, -0.8 mm. 500nl of the diluted lentivirus (1/3 dilution in saline solution of the KIF1A shRNA or the Scr shRNA) at 0.5µl/min speed using a nanoinjector. The capillary was slowly removed 1 min after the end of the injection to prevent the leak of the injected solution. The skull was then washed with saline solution, the skin was sutured and 1ml of NaCl 0.9% was injected subcutaneously. After surgery, mice were put alone in a warmed cage and monitored daily throughout recovery.

## Plasmids

The VAMP2-mCherry construct was a kind gift from T Ryan's laboratory. *Vglut1* cDNA sequence was amplified from an adult mouse brain. Its sequence from the 104th amino acid to the end was cloned after the pHluorin sequence in a SmaI site in a superecliptic pHluorin containing vector used in *Fernandez-Alfonso and Ryan, 2008*. KIF1A shRNA construct (JL-35, target sequence: GACCGGAC CTTCTACCAGT) has already been published (*Kevenaar et al., 2016*). It has been inserted in an EGFP-pSuper vector between NdeI and PstI sites. The control Scramble shRNA is a mouse universal scramble obtained from the scrambled order of HIF-1α nucleotides (target sequence: GGGTGAAC TCACGTCAGAA) (*Yu et al., 2004*). It has been inserted in an EGFP-pSuper vector between NdeI and

PstI sites. BDNF-mCherry construct and lentivirus, previously published (*Hinckelmann et al., 2016*), have been used for axonal transport experiments.

For lentivirus production, all the plasmids were cloned into a pSin vector (*Drouet et al., 2009*) by Gateway system (Life Technology) at the GIN virus production facility as described before (*Bruyère et al., 2020*). VAMP2-mCherry, sh-Kif1a, and sh-Scr lentiviruses were produced by the ENS Lyon Vectorology Facility.

## Microfluidic fabrication

Microfluidic devices were generated as previously described (*Lenoir et al., 2021*; *Virlogeux et al., 2018*). Briefly, we modified the size of the microchannels (3 µm width, 3 µm height, and 500 µm length) of polydimethylsiloxane microfluidic device (*Taylor et al., 2005*). After amplification and production, microfluidic devices were sealed on Iwaki boxes using plasma cleaner. The upper chamber was then coated with poly-D-lysin (0.1 mg/ml) and the lower chamber was coated with poly-D-lysin (0.1 mg/ml) and laminin (10 µg/ml). After overnight incubation at 4°C, microfluidic devices were washed two times with the growing medium. Microchambers were then placed in the incubator until neurons were plated.

## Videomicroscopy

Videorecording of neurons plated in microfluidic devices was performed at DIV 12. Before recordings, DIV 12 neurons in the microchamber were carefully inspected and selected based on the absence of cell contamination. For double transductions (with sh-RNA lentiviruses), the transport of mCherry-tagged cargo was analyzed within GFP-positive axons. Images were acquired every 200 ms for 1 min on an inverted microscope (Axio Observer, Zeiss) with X63 oil-immersion objective (1.46NA) coupled to a spinning-disk confocal system (CSU-W1-T3; Yokogawa) connected to an electron-multiplying CCD (charge-coupled device) camera (ProEM+1024, Princeton Instrument) at 37°C and 5% $CO_2$.

For the study of the exocytosis events, images were acquired every 200 ms for 1 min on an inverted microscope (Axio Observer, Zeiss) with X63 oil-immersion objective (1.46NA) coupled to a spinning-disk confocal system (CSU-W1-T3; Yokogawa) with TIRF microscopy (Nikon/Roper, Eclipse Ti) equipped with a camera Prime 95B sCMOS (Telelyne Photometrics) at 37°C and 5% $CO_2$. The same three fields per microchambers were acquired before and after a 4AP-bicuculline (respectively 2.5 mM and 50 µM) stimulation of the presynaptic chamber, four times in total (one before and three after stimulation).

## Immunostaining

Neurons from the reconstituted corticostriatal network were fixed with a PFA/sucrose solution (4%/4% in PBS) for 20 min at room temperature (RT). After three washes of PBS, neurons were incubated first with a blocking solution (BSA 1%, NGS 2%, Triton X-100 0.1%) and then with primary antibodies for KIF1A (Abcam, #ab180153, 1:100, rabbit), HTT (Millipore, #MAB2166, 1:500, mouse), and mCherry (Fisher Scientific, #16D7, 1:200, rat) overnight at 4°C. The next day, neurons were washed three times with PBS followed by 1 hr incubation at RT of appropriate secondary antibodies (1:1000) and finally washed again three times with PBS. Images were acquired with a X63 oil-immersion objective (1.4 NA) using an inverted confocal microscope (LSM 710, Zeiss) coupled to an Airyscan detector. For 2D-STED microscopy, we used the Abberior kit containing the secondary antibodies (STAR RED anti-mouse or rabbit, STAR ORANGE anti-mouse or rabbit) and coverslips were mounted with the Abberior mount solid. Images were taken with a 100× oil-immersion objective (1.46 NA) using the Abberior 2D-STEDYCON upright confocal microscope.

For brain slices, brains were incubated in PFA 4% overnight and washed with PBS three times the next day. Then, brains were cut into 100-µm-thick slices using a vibratome. The slices were incubated with a blocking solution (0.3% Triton, 10%NGS in PBS) for 2 hr at RT and then with antibody against GFP (Institut Curie, A-P-R#06) overnight at 4°C. The day after, the primary antibody was removed by three washes of PBS before incubating the slices with the associated secondary antibody and finally with three washes of PBS. Finally, slices were incubated with DAPI (1/4000) for 15 min, washed three times with PBS, mounted on Superfrost slides by using Dako Faramount Aqueous Mounting Medium solution and coverslips. The slices were acquired with a ×10 objective (0.45 NA) using a slide scanner (AxioScan Z1, Zeiss) and with a ×10 objective (0.3 NA) using an inverted confocal microscope

(LSM 710, Zeiss) coupled to an Airyscan detector to improve signal-to-noise ratio and to increase the resolution.

### In situ PLA

The NaveniFlex MR kit (Navinci #NF.MR.100) was used to study the interaction of endogenous HTT with endogenous KIF1A/VAMP2 in cortical neurons. The assay was performed following the manufacturer's instructions. Primary antibodies were incubated with the same dilutions used for immunocytochemistry experiments. Images were acquired with a X63 oil-immersion objective (1.4 NA) using an inverted confocal microscope (LSM 710, Zeiss).

### Western blotting

Cortical neurons were plated in free culture, transduced at DIV 1 with sh-Kif1a or Sh-scr, and lysed at DIV 5 in NetN buffer (20 mM Tris-HCl pH 8, 120 mM NaCl, 1 mM EDTA, 0.5% NP40) complemented with protease inhibitor cocktail (Roche).

A vesicular fraction from brains was prepared as described in *Hinckelmann et al., 2016*. Briefly, brains were homogenized in lysis buffer (10 mM HEPES-KOH, 175 mM L-aspartic acid, 65 mM taurine, 85 mM betaine, 25 mM glycine, 6.5 mM $MgCl_2$, 5 mM EGTA, 0.5 mM D-glucose, 1.5 mM $CaCl_2$, 20 mM DTT pH 7.2, protease inhibitor from Roche) on ice with a glass potter and then with a 25 G needle. Lysates were then centrifuged (12,000 RPM) and the supernatant, considered as the total fraction, is then centrifuged (3000 RPM for 10 min). The resulting supernatant was centrifuged (12,000 RCF for 40 min). The supernatant was then ultracentrifuged (100,000 × $g$) to obtain the vesicular fraction (the pellet) and the cytosolic fraction (the supernatant).

All types of lysed samples were dosed by a Bradford reagent to quantify the protein concentration and then analyzed by SDS-PAGE transferred to PVDF membranes. Then, membranes were incubated for 45 min in a 5% BSA TBST (10 mM Tris pH 8, 150 mM NaCl, 0.5% Tween 20) solution and incubated with primary antibodies against KIF1A (Abcam #ab180153, 1:5000), VAMP2 (Synaptic Systems #104211, 1:1000), pS421 (Chemicon #AB9562, 1:500), Vinculin (Sigma #V9131, 1:10,000), p150 (BD laboratories, #612708, 1:1000), Tubulin (Sigma #T9026, 1:1000) at 4°C, overnight. The next day, membranes were washed at least three times with TBST and incubated with secondary antibodies conjugated with horseradish peroxidase against mouse or rabbit (1:1000) for 2 hr at RT. Membranes were finally revealed with ECL (Thermo Scientific) after three washes of TBST.

### Electron microscopy

We anesthetized 3- to 4-month-old animals with 1 ml/kg of Doléthal and perfused them transcardially with cold PBS followed by 2% paraformaldehyde, 2% glutaraldehyde, and 0.1 M cacodylate cold solution. We removed brains from the skull and fixed them in a 0.1 M phosphate buffer pH 7.2 with 2% of glutaraldehyde and 2% of paraformaldehyde for 48 hr at 4°C before obtaining 2-mm-thick or 100-μm-thick slices from a mold and a vibratome, respectively. A 1 mm square piece of tissue was removed from the DLS; samples were then fixed again with the same solution for 72 hr, washed with phosphate buffer, and then post-fixed in a 0.1 M phosphate buffer pH 7.2 with 1% osmium tetroxide for 1 hr at 4°C. After extensive washes with water, samples were then stained with a solution of 1% uranyl acetate pH 4 in water for 1 hr at 4°C. They were further dehydrated through a gradient of ethanol (30%–60%–90% and three at 100%) and infiltrated with a solution of 1/1 epon/alcohol 100% for 1 hr and followed by several baths of fresh epon (Fukka) for 3 hr. The resin was then poured into capsules containing the samples, heated at 60°C for 72 hr for polymerization, and finally cut into ultrathin sections with an ultramicrotome (Leica). Sample sections were then post-stained with fresh solutions of 5% uranyl acetate and 0.4% of lead citrate, observed with a transmission electron microscope at 80 kV (JEOL 1200EX) and images were acquired with a digital camera (Veleta, SIS, Olympus). Analysis was performed with ImageJ and quantification of the number of synapses was performed on axon-free neuropil regions (*Zhang et al., 2015*).

### Brain slice preparation and whole-cell patch-clamp recordings

All experiments were performed in accordance with the guidelines of the local animal welfare committee (Center for Interdisciplinary Research in Biology Ethics Committee) and the EU (directive

2010/63/EU). We prepared horizontal brain slices containing the somatosensory S2 cortex and the corresponding corticostriatal projection field in the dorsal striatum from mice (2–3 months of age) using a vibrating blade microtome (VT1200S, Leica Micosystems, Nussloch, Germany). Brains were sliced in a 5% $CO_2$/95% $O_2$-bubbled, ice-cold cutting solution containing (in mM): 125 NaCl, 2.5 KCl, 25 glucose, 25 $NaHCO_3$, 1.25 $NaH_2PO_4$, 2 $CaCl_2$, 1 $MgCl_2$, and 1 pyruvic acid, and then transferred into the same solution at 34°C for 60 min and then moved to RT.

For whole-cell patch-clamp recordings, borosilicate glass pipettes of 4–6 MΩ resistance contained (in mM): 105 K-gluconate, 30 KCl, 10 HEPES, 10 phosphocreatine, 4 ATP-Mg, 0.3 GTP-Na, 0.3 EGTA (adjusted to pH 7.35 with KOH). The composition of the extracellular solution was (in mM): 125 NaCl, 2.5 KCl, 25 glucose, 25 $NaHCO_3$, 1.25 $NaH_2PO_4$, 2 $CaCl_2$, 1 $MgCl_2$, and 10 µM pyruvic acid, bubbled with 95% $O_2$ and 5% $CO_2$. Signals were amplified using EPC10-2 amplifiers (HEKA Elektronik, Lambrecht, Germany). All recordings were performed at 34°C using a temperature control system (Bath-controller V, Luigs & Neumann, Ratingen, Germany) and slices were continuously superfused at 2–3 ml/min with the extracellular solution. Slices were visualized on an Olympus BX51WI microscope (Olympus, Rungis, France) using a 4×/0.13 objective for the placement of the stimulating electrode and a 40×/0.80 water-immersion objective for localizing cells for whole-cell recordings. The series resistance was not compensated. Recordings were sampled at 10 kHz, using the Patchmaster v2x32 program (HEKA Elektronik).

For paired-pulse protocols, electrical stimulations were performed with a bipolar electrode (Phymep, Paris, France) placed in the layer V of the somatosensory S2 cortex. Electrical stimulations were monophasic at constant current (ISO-Flex stimulator, AMPI, Jerusalem, Israel). Currents were adjusted to evoke 50–200 pA EPSCs. Repetitive control stimuli were applied at 0.1 Hz. For each ISI, 20 successive EPSCs were individually measured and then averaged. Variation of input and series resistances above 20% led to the rejection of the experiment. Off-line analysis was performed with Fitmaster (Heka Elektronik). Statistical analysis was performed with Prism 5.02 software (San Diego, CA, USA). All results are expressed as mean ± SEM. Statistical significance was assessed in non-parametric Mann-Whitney, one-sample t-tests using the indicated significance threshold (p).

## Mass spectrometry

This analysis follows that of *Migazzi et al., 2021*. Briefly, vesicular fraction from brains obtained as described earlier was first pre-cleared for an hour at 4°C with protein A Sepharose beads (Sigma-Aldrich-P9424) and then immunoprecipitated for 3 hr at 4°C by agarose beads preincubated with rabbit anti-HTT D7F7 antibody (Cell Signaling, Cat#5656). To remove the non-specific binding, the beads were washed three times with the lysis buffer and bound proteins are finally eluted with Laemmli buffer. The HTT corresponding band on the western blot was cut and analyzed. MS was performed with an LTQ Orbitrap XL mass spectrometer (Thermo Scientific), equipped with a nanoESI source (Proxeon). The top eight peaks in the mass spectra (Orbitrap; resolution, 60,000) were selected for fragmentation (CID; normalized collision energy, 35%; activation time, 30 ms, q-value, 0.25). Proteins were identified using the MaxQuant software package version 1.2.2.5 (MPI for Biochemistry, Germany) and UniProt database version 04/2013.

## Quantification and statistical analyses

### Transport analysis

Vesicle velocity, directional flow, and vesicle number were measured on 100 µm of neurite using KymoTool Box ImageJ plugin, as previously described (*Virlogeux et al., 2018*). Anterograde or retrograde speeds describe, respectively, the mean speed of anterograde or retrograde segmental movement of a vesicle. Static vesicles are those without any movement during the recording. Linear flow and directionality were calculated as in *Virlogeux et al., 2018*.

### Electrophysiology analysis

For each ISI, 20 successive EPSCs were individually measured and then averaged. Variation of input and series resistances above 20% led to the rejection of the experiment. Off-line analysis was performed with Fitmaster (Heka Elektronik). Statistical analysis was performed with Prism 5.02 software (San Diego, CA, USA). All results are expressed as mean ± SEM. Statistical significance was assessed in non-parametric Mann-Whitney, one-sample t-tests using the indicated significance threshold (p).

## Immunostaining

Immunostained vesicles in distal axons were quantified as previously shown (*Scaramuzzino et al., 2022*). Briefly, we used a customized macro for ImageJ where the images are enhanced using a DoG filter adapted to the vesicle size. Masks are created on each channel using manual thresholding that is kept constant for each individual channel and replicates. Finally, the number of particles is automatically counted for the single and dual channels and expressed as the percentage of colocalization. For PLA analysis, the number of fluorescent dots in 100 µm$^2$ was normalized on the total number of nuclei (DAPI$^+$).

## Exocytosis events

The same three fields per microchamber were acquired before and after a 4AP-bicuculline (respectively 2.5 mM and 50 µM) stimulation of the presynaptic chamber, four times in total (one before and three after stimulation). The movies were analyzed using a customized macro for ImageJ, thus the recording of the amplitude and the number of exocytosis events were automatized. The number of events was expressed as follows:

$$y_i = \frac{\left(y_{i,post} - y_{i,pre}\right) - \frac{\sum x_j}{n}}{100} + 100$$

where

- $y_i$ expresses the difference of the number of events after ($y_{i,\,post}$) minus before ($y_{i,\,pre}$) stimulation in HTT-SD neurons of the field $i$,
- $x_j$ expresses the difference of the number of events after minus before stimulation in WT neurons of the field j and n of them have been averaged.
- the final value was normalized to 1, that is a given HTT-SD neuron field, whose activity after stimulation increased as much as that of the average of the WT neuron fields, will display a value of 1.
- The amplitude of the signal from stimulated neurons was normalized by that of the same neuron before stimulation.

## Electron microscopy analysis

We used ImageJ to analyze synapse morphology. We counted the number of synapses in axon-free neuropil regions (*Zhang et al., 2015*). SVs were numbered according to their physical features (size, gray scale, and shape). The presynaptic zone, which contains the active zone, was defined as the zone facing the PSD. The 40-nm-wide zones in the presynapse were defined according to their location relative to the active zone.

## Statistical analysis

Statistical calculations were performed using GraphPad Prism 6.0. Statistical parameters (replication, sample size, SEM, etc.) are reported in the figure legends. For each dataset, we identified outliers using the ROUT test (Q=1%) and removed them from analysis. We performed a Shapiro-Wilk normality test with the threshold set at α=0.05; if the data followed a normal distribution, we used parametric tests, and if not, we used non-parametric tests. If we were analyzing two conditions we used a t-test (or a Mann-Whitney test if non-parametric). If comparing more than two conditions, we used a one-way ANOVA followed by Tukey's post hoc analysis (or a Kruskal-Wallis test followed by Dunn's post hoc analysis if non-parametric). If the datasets were interdependent, we used a two-way ANOVA followed by Tukey's post hoc analysis if more than two groups are compared, or a Sidak's post hoc analysis if only two groups are analyzed. For a nonlinear fit, we did a run test to determine whether the curve deviates systematically from the data. Low p value (ns) indicates that the curve poorly describes the data. *p<0.05; **p<0.01; ***p0.001; ****p<0.0001; ns, non-significant.

## Acknowledgements

We thank Sandrine Humbert, Sebastien Carnicella, Alain Marty, members of the Saudou, Humbert, and Venance labs for comments; Vicky Brandt for critical editing; Béatrice Blot, Aurélie Genoux,

Nagham Badreddine, and Camille Brodier for technical help; T Ryan for the gift of pHluorin plasmid; C Hoogenraad for the gift of sh-Kif1a plasmid; Y Saoudi for help with image acquisitions and the Photonic Imaging Center of Grenoble Institut Neuroscience (PIC-GIN) which is part of the ISdV core facility and is certified by the IBiSA label, Karin Pernet-Gallay and Anne Bertrand for help with electron microscopy. We acknowledge the contribution of Gisèle Froment, Didier Nègre, and Caroline Costa and the AniRA lentivectors production facility from the CELPHEDIA Infrastructure and SFR Biosciences (UAR 3444/CNRS, US8/Inserm, ENS de Lyon, UCBL). This work was supported by grants from the European Research Council (ERC) under the European Union's Horizon 2020 research and innovation program AdG grant agreement no. 834317, Fueling Tranport (FS); Agence Nationale de la Recherche: ANR-15-IDEX-02 NeuroCoG (FS) in the framework of the 'Investissements d'avenir' program; ANR-18-CE16-0009-01 AXYON (FS); Fondation pour la Recherche Médicale, FRM, DEI20151234418 (FS) and AGEMED program from INSERM (FS). The Saudou laboratory is part of the Grenoble Center of Excellence in Neurodegeneration (GREEN). CS was supported by a Postdoctoral fellowship from: Fondation pour la Recherche Médicale, FRM (SPF20140129405) and EMBO LTF (ALTF 693-2015). HV was supported by a PhD fellowship from Association Huntington France and by a fellowship from Fondation pour la Recherche Médicale, FRM (FDT201904008035). The illustrations were created with BioRender.com (License # WY253PKO7J, # HY253PJ859, # PU253PJZFW, # QX253PKGGZ).

## Additional information

### Funding

| Funder | Grant reference number | Author |
|---|---|---|
| Horizon 2020 - Research and Innovation Framework Programme | 834317 | Frédéric Saudou |
| Agence Nationale de la Recherche | ANR-15-IDEX-02 NeuroCoG | Frédéric Saudou |
| Agence Nationale de la Recherche | ANR-18-CE16-0009-01 AXYON | Frédéric Saudou |
| Fondation pour la Recherche Médicale | FRM DEI20151234418 | Frédéric Saudou |
| Fondation pour la Recherche Médicale | SPF20140129405 | Chiara Scaramuzzino |
| European Molecular Biology Organization | ALTF 693-2015 | Chiara Scaramuzzino |
| Association Huntington France | PhD fellowship | Hélène Vitet |
| Fondation pour la Recherche Médicale | FDT201904008035 | Hélène Vitet |

The funders had no role in study design, data collection and interpretation, or the decision to submit the work for publication.

### Author contributions

Hélène Vitet, Conceptualization, Funding acquisition, Investigation, Methodology, Writing – original draft, Writing – review and editing; Julie Bruyère, Conceptualization, Supervision, Investigation, Methodology, Writing – review and editing; Hao Xu, Yah-Sé Abada, Investigation, Methodology, Writing – review and editing; Claire Séris, Investigation, Methodology; Jacques Brocard, Methodology, Writing – review and editing; Benoît Delatour, Supervision, Writing – review and editing; Chiara Scaramuzzino, Conceptualization, Supervision, Funding acquisition, Investigation, Methodology, Writing – original draft, Writing – review and editing; Laurent Venance, Conceptualization, Supervision, Writing – original draft, Writing – review and editing; Frédéric Saudou, Conceptualization, Supervision, Funding acquisition, Writing – original draft, Writing – review and editing

## Author ORCIDs
Jacques Brocard http://orcid.org/0000-0002-0752-5737
Chiara Scaramuzzino http://orcid.org/0000-0001-9454-8701
Laurent Venance http://orcid.org/0000-0003-0738-1662
Frédéric Saudou http://orcid.org/0000-0001-6107-1046

## Ethics

All experimental procedures were performed in an authorized establishment (Grenoble Institut Neurosciences, INSERM U1216, license #B3851610008) in strict accordance with the directive of the European Community (63/2010/EU). The project was approved by the French Ethical Committee (Authorization number: APAFIS#18126-2018103018299125 v2) for care and use of laboratory animals and performed under the supervision of authorized investigators.

## Decision letter and Author response

Decision letter https://doi.org/10.7554/eLife.81011.sa1
Author response https://doi.org/10.7554/eLife.81011.sa2

## Additional files

### Supplementary files
• MDAR checklist

### Data availability
All datasets generated and analyzed during the study are included in the manuscript and in the supporting files. Source data files have been provided for Figure 1, Figure 1—figure supplement 1, Figure 2, Figure 2—figure supplement 2, Figure 3, Figure 3—figure supplement 3, Figure 4, Figure 4—figure supplement 4, Figure 5, Figure 5—figure supplement 5, Figure 6, Figure 6—figure supplement 6, Figure 6—figure supplement 7, Figure 7, and Figure 8.

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
