## [Editor Report]

This important study provides solid in vitro and in vivo data supporting the connection between Huntingtin's (HTT) phosphorylation state and the recruitment of Kif1A in the axonal anterograde trafficking of synaptic vesicles precursors. This works highlights the role of HTT-Kif1A synergy in modulating synaptic vesicle precursor transport and synaptic function, adding important information related to how synaptic vesicle pools are formed and replenished.

---

## [Decision Letter]

**Decision letter after peer review:**

Thank you for submitting your article "Huntingtin-KIF1A-mediated axonal transport of synaptic vesicle precursors influences synaptic transmission and motor skill learning in mice" for consideration by *eLife*. Your article has been reviewed by 2 peer reviewers, and the evaluation has been overseen by a Reviewing Editor and Suzanne Pfeffer as the Senior Editor. The reviewers have opted to remain anonymous.

Essential revisions:

1. The authors conclude that there is a change in synaptic transmission based on subtle modulation of total synaptic vesicle number. However, it is not very well supported, and the literature argues against total vesicle number being a major point of control for neurotransmission. The authors should discuss this point and take into account the direct effect on vesicle exocytosis or short-term synaptic plasticity.

2. Figure 1A-B would be strengthened by proximity ligation assays showing the extent of endogenous interaction between huntingtin and VAMP2 in axons of WT versus HTT-SD and HTT-SA mutants. Perhaps this can be incorporated in figure 5B-C, where to this reviewer's opinion, a PLA approach would be better to show and quantify the interaction between KIF1A and HTT (see the separate point). Alternatively, a PLA for KIF1A and HTT combined with IF for VAMP could work as well.

3. Figure 1C – How do the authors explain the high variance in the HTT-SA mutant? It seems that it could reach significance with 1-2 more devices. Why do they think they do not see reduced synaptic release but a trend toward increased one? Representative movies would help understand the system.

4. Figure 4 – Do the authors detect a change also in synapse size? The synaptic vesicle count is not normalized to synapse area/size/volume.

5. Figure 5 – The images in figures 5B-C are challenging to interpret and, therefore, also fail to represent the data. We don't see much colocalization. VAMP2 does not seem to colocalize with Kif1A, while it seems to overlap better with HTT. Could the authors provide a quantification of Kif1a-VAMP2 and HTT-VAMP2 colocalization and discuss the implications? As mentioned before, a proximity ligation assay would be suitable for strengthening this point. Also, since this experiment is done on fixed cells, the authors should use cells that were not-infected with VAMP2-mCherry cells but rather observe the endogenous VAMP2 colocalization in KIF1A and HTT. It will be nice to show also VAMP2 levels in 5D-E.

6. The statistical approach should be better substantiated. The authors often make use of the Man Whitney test, which as far as I know is non-parametric and does not assume normal distribution. However, the data presented are always averages with standard error of the means (s.e.m.). Seems to me to be a contradiction. The authors should comment on why they chose this test over a parametric one and, if their dataset is not normally distributed, maybe they should check if their representation is correct. In addition, ANOVA is used with different post-tests. Most of the time Dunn is used, but also Sidak and Tuckey are used. I could not infer from the dataset why that is and maybe the authors could explain.

– In Figure 1C the significance was determined using a t-test? ANOVA seems more appropriate. Which statistical assay was used in Figure 3?

– Figure 1A and B. Is the comparison between HTT-SD and HTT-SA significant?

– Figure 2B. All the latency to fall of all the mice in all the repeat for all the days tested have been pulled together? If this is case, this representation of the rotarod data is unusual, please explain what the advantage is compared to just showing the curve of the entire experiment as done in panel C.

– Figure 3C. The quantification has an abundance of asterisks. It is unclear what was compared with what.

– Fig S2. Why are HTT-SD and HTT-SA separately compared to the wild type control? They should be grouped together in the same chart and ANOVA should be used.

7. Why the authors do not elaborate more on the difference or lack thereof between HTT-SD and HTT-SA and they are so quick to drop the HTT-SA mutant from the rest of study? The alanine mutant are expected to be non phosphorylatable and thus should present a more drastic difference in relation to the constitutive phosphorylated mutant one (SD). The wild type HTT, which can still be phosphorylated, should be somewhere in the middle. This seems counterintuitive. Have the authors tested for the phosphorylation status of their mutants?

8. It seems counterintuitive that HTT-SD promotes SVPs transport, but results in a behavioral deficit. Although the authors provide the explanation of these results, but it would be helpful if the authors could provide more evidence. For example, the hypothesis of an increase in docked SVs could be tested. The authors provide already EM images (Figure 4 and 7B). Could they quantify the number of docked vesicles? If this is difficult, maybe the number of vesicles adjacent to the membrane?

*Reviewer #1 (Recommendations for the authors):*

I find this manuscript suitable for publication following revisions and corrections as detailed below.

1. Figure 1A-B would be strengthened by proximity ligation assays showing the extent of endogenous interaction between huntingtin and VAMP2 in axons of WT versus HTT-SD and HTT-SA mutants. Perhaps this can be incorporated in figure 5B-C, where to this reviewer's opinion, a PLA approach would be better to show and quantify the interaction between KIF1A and HTT (see the separate point). Alternatively, a PLA for KIF1A and HTT combined with IF for VAMP could work as well.

2. Figure 1C – How do the authors explain the high variance in the HTT-SA mutant? It seems that it could reach significance with 1-2 more devices. Why do they think they do not see reduced synaptic release but a trend toward increased one? Representative movies would help understand the system.

3. Figure 4 – Do the authors detect a change also in synapse size? The synaptic vesicle count is not normalized to synapse area/size/volume.

4. Figure 5 – The images in figures 5B-C are challenging to interpret and, therefore, also fail to represent the data. As mentioned before, a proximity ligation assay would be suitable for strengthening this point. Also, since this experiment is done on fixed cells, the authors should use cells that were not-infected with VAMP2-mCherry cells but rather observe the endogenous VAMP2 colocalization in KIF1A and HTT. It will be nice to show also VAMP2 levels in 5D-E.

*Reviewer #2 (Recommendations for the authors):*

The manuscript presents a solid set of data, which are in my opinion broadly supportive of the authors conclusions. A lot of work has gone into the incorporation of multiple approaches. I have a set of minor considerations that I hope might be helpful to the authors.

1) The statistical approach should be better substantiated. The authors often make use of the Man Whitney test, which as far as I know is non-parametric and does not assume normal distribution. However, the data presented are always averages with standard error of the means (s.e.m.). Seems to me to be a contradiction. The authors should comment why they chose this test over a parametric one and, if their dataset is not normally distributed, maybe they should check if their representation is correct. In addition, ANOVA is used with different post-tests. Most of the time Dunn is used, but also Sidak and Tuckey are used. I could not infer from the dataset why that is and maybe the authors could explain.

In Figure 1C the significance was determined using a t-test? ANOVA seems more appropriate. Which statistical assay was used in Figure 3?

2) Figure 1A and B. Is the comparison between HTT-SD and HTT-SA significant?

3) Figure 2B. I am not sure if I interpret this chart correctly. All the latency to fall of all the mice in all the repeat for all the days tested have been pulled together? If this is case, I am not sure I have encountered this representation of the rotarod data before and I wonder what the advantage is compared to just showing the curve of the entire experiment as done in panel C.

4) Figure 3C. The quantification has an abundance of asterisks. I could not figure out what was compared with what. I wonder if the comparisons could be expressed more clearly.

5) Figure 5C. Maybe the author could find another image. I cannot see much colocalization. VAMP2 does not seem to colocalize with Kif1A, while it seems to overlap better with HTT. Could the authors provide a quantification of Kif1a-VAMP2 and HTT-VAMP2 colocalization and discuss the implications?

6) Fig S2. Why are HTT-SD and HTT-SA separately compared to the wild type control? They should be grouped together in the same chart and ANOVA should be used.

7) Following from point 6 and 2, I am not sure why the authors do not elaborate more on the difference or lack thereof between HTT-SD and HTT-SA and they are so quick to drop the HTT-SA mutant from the rest of study. I expected that the alanine mutant would be non phosphorylatable and thus should present a more drastic difference in relation to the constitutive phosphorylated mutant one (SD). The wild type HTT, which can still be phosphorylated, should be somewhere in the middle. This seems counterintuitive. Have the authors tested for the phosphorylation status of their mutants?

8) Seems counterintuitive that HTT-SD promotes SVPs transport, but results in a behavioral deficit. I understand the explanation provided, but I wonder if the authors could provide more evidence. For example, the hypothesis of an increase in docked SVs could be tested. The authors provide already EM images (Figure 4 and 7B). Could they quantify the number of docked vesicles? If this is difficult, maybe the number of vesicles adjacent to the membrane?

---

## [Author Response]

Essential revisions:1. The authors conclude that there is a change in synaptic transmission based on subtle modulation of total synaptic vesicle number. However, it is not very well supported, and the literature argues against total vesicle number being a major point of control for neurotransmission. The authors should discuss this point and take into account the direct effect on vesicle exocytosis or short-term synaptic plasticity.

There are two issues here: the reviewer is not convinced by our data that synaptic transmission is affected by subtle changes in vesicle number, and believes the literature argues against vesicle number having much influence on neurotransmission. We'll take each issue in turn.

We decided to analyze the distribution of vesicles in the presynaptic compartment. We divided the PSD into three 40 nm-wide zones: the proximal, active zone 1 (0 to 40 nm); zone 2 (40 to 80 nm); and the distal zone 3 (80 to 120 nm). Although vesicle pools are not anatomically segregated (Rizzoli, 2014), the most proximal zone is likely to contain the readily releasable pool (RRP) while the most distal zone is likely to be enriched in vesicles from the reserve pool (RP) (new Figure 4C). WT and HTT-SD synapses did not differ in these last 120 µm of the axon terminal area (new Figure 4 – supplement 1A), but HTT-SD presynapses contained significantly more vesicles in zone 3 (p<0.001; new Figure 4D). This correlates with the increased anterograde axonal transport leading to an accumulation of vesicles at the synapse, adding to the already-filled reserve pool.

The literature supports our contention that this greater number of vesicles in the distal pool might influence the replenishment of the RRP (Kidokoro et al., 2004) and/or the recycling pool (Ratnayaka et al., 2012). Several studies have linked the reserve pool to neurotransmission: In *Drosophila*, reduced recruitment of SVs from the reserve pool to the RRP impairs short-term synaptic plasticity and memory (Kidokoro et al., 2004). While location relative to the membrane does not reliably define vesicular pool identity (Denker, Krohnert, and Rizzoli, 2009; Fernandez-Alfonso and Ryan, 2008; Fredj and Burrone, 2009), our results are in agreement with the increased anterograde axonal transport of SVP we observed in vitro in HTT-SD neurons. Indeed, it is expected that vesicles are likely to accumulate mostly where the SVPs detach from the KIF1A motor, a location which is away from the membrane and the active zone (Bodaleo and Gonzalez-Billault, 2016). In addition, newly produced vesicles arriving at the presynaptic zone are preferentially released (Truckenbrodt et al., 2018). Since all the vesicle pools are dynamic and interdependent (Chamberland and Toth, 2016; Chanaday, Cousin, Milosevic, Watanabe, and Morgan, 2019), this study suggests that modifying the distal pool of vesicles would affect this dynamic instead of the number of docked vesicles adjacent to the membrane. This hypothesis is supported by the fact that modulating the size or dynamicity [?] of the reserve pool affects plasticity and memory (Corradi et al., 2008; De Rossi et al., 2020; Gitler, Cheng, Greengard, and Augustine, 2008; Skorobogatko et al., 2014).

These findings are introduced in the Results section and now addressed in the discussion.

2. Figure 1A-B would be strengthened by proximity ligation assays showing the extent of endogenous interaction between huntingtin and VAMP2 in axons of WT versus HTT-SD and HTT-SA mutants. Perhaps this can be incorporated in figure 5B-C, where to this reviewer's opinion, a PLA approach would be better to show and quantify the interaction between KIF1A and HTT (see the separate point). Alternatively, a PLA for KIF1A and HTT combined with IF for VAMP could work as well.

Good suggestion. We now provide PLA experiments and show a significantly increased association of HTT with KIF1A in HTT-SD neurons (new Figure 5D and results). We were unsuccessful in combining IF for VAMP with PLA for KIF1A and HTT, but we measured association of KIF1A or HTT to VAMP2 vesicles in HTT-SD neurons using PLA (new Figure 5 —figure supplement 1C-D) and Airyscan microscopy (new Figure 5 —figure supplement 1A). These latter studies did not find consistent changes in the association of HTT or KIF1A to VAMP2. See also our response under point 5 below.

3. Figure 1C – How do the authors explain the high variance in the HTT-SA mutant? It seems that it could reach significance with 1-2 more devices. Why do they think they do not see reduced synaptic release but a trend toward increased one? Representative movies would help understand the system.

This question prompted us to re-analyze the original data and plot all the regions of interest (ROI) values rather than their mean as we did before. We analyzed 6712 events for WT neurons, 4640 events for the SD neurons and 5176 for the SA neurons within 60 seconds. This increased the statistical power and, because the data do not follow a normal distribution, we presented the data as a box-whisker plot (new Figure 1C). Although one point is particularly low (but was not an outlier), the distribution is now seen to be very similar to the one in HTT-SD condition. There does not appear to be a trend toward increased synaptic release.

To help the reader to better understand our experimental set up, we now provide one movie (after stimulation) in which the distal part of the axonal channels can be seen in the upper part of the frame, the synaptic chamber in the middle and the distal part of the dendritic channels in the lower part of the frame. Arrows indicate transient spots of exocytosis that occur in the synaptic compartment and that have been quantified using a customized macro for Image, as described now in the Material and methods section (new Video 1).

4. Figure 4 – Do the authors detect a change also in synapse size? The synaptic vesicle count is not normalized to synapse area/size/volume.

Excellent point. We analyzed the images obtained from electron microscopy and found that indeed, HTT-SD presynapses are smaller than WT (new Figure 4Biii). Normalizing the synaptic vesicle count to synapse size confirms that HTT-SD synapses have more vesicles than WT synapses (new Figure 4Biv).

5. Figure 5 – The images in figures 5B-C are challenging to interpret and, therefore, also fail to represent the data. We don't see much colocalization. VAMP2 does not seem to colocalize with Kif1A, while it seems to overlap better with HTT. Could the authors provide a quantification of Kif1a-VAMP2 and HTT-VAMP2 colocalization and discuss the implications? As mentioned before, a proximity ligation assay would be suitable for strengthening this point. Also, since this experiment is done on fixed cells, the authors should use cells that were not-infected with VAMP2-mCherry cells but rather observe the endogenous VAMP2 colocalization in KIF1A and HTT. It will be nice to show also VAMP2 levels in 5D-E.

As explained above under point 2, we now provide PLA experiments demonstrating a specific interaction between HTT and KIF1A and a statistically significant increased association of HTT with KIF1A in HTT-SD neurons (new Figure 5D). We also performed high-resolution Airy Scan confocal experiments to quantify HTT interaction with KIF1A (new Figure 5C) and found a significant increase in the interaction between HTT and KIF1A in HTT-SD neurons.

As requested, we also performed PLA experiments on endogenous VAMP2 colocalization with KIF1A and HTT (new Figure 5 —figure supplement 1B-C), but we did not observe significant changes in the association of HTT or KIF1A to endogenous VAMP2. Using Airy Scan confocal microscopy (new Figure 5 —figure supplement 1A) we observed similar association of HTT or KIF1A to VAMP2-mCherry. This suggests that most of the effect induced by HTT phosphorylation is through the increased interaction between motors (KIF1A) and HTT. These findings on the localization of HTT on vesicles independently of its phosphorylation at S421 is in agreement with our previous studies (Bruyere et al., 2020; Colin et al., 2008). Finally, we also show VAMP2 levels (new Figure 5E and Figure 5 —figure supplement 1D with new quantifications).

Together with our proteomic data, our results obtained with 2D-STED microscopy, PLA, Airy Scan microscopy, and biochemical analyses of synaptic vesicle preparation (new Figure 5E) provide strong evidence that KIF1A interacts with HTT on VAMP2 immuno-positive vesicles and that the HTT-KIF1A interaction is enhanced in HTT-SD neurons.

6. The statistical approach should be better substantiated. The authors often make use of the Man Whitney test, which as far as I know is non-parametric and does not assume normal distribution. However, the data presented are always averages with standard error of the means (s.e.m.). Seems to me to be a contradiction. The authors should comment on why they chose this test over a parametric one and, if their dataset is not normally distributed, maybe they should check if their representation is correct.

We apologize for not better describing the statistics, and we now provide more information in both the Methods and the figure legends. We checked the normal distribution with a Shapiro-Wilk test. In most of the cases, our data sets did not follow a normal distribution. This is the reason why we chose a non-parametric test to compare 2 conditions. Then, as suggested by the reviewers, we modified the representation of the data that were analyzed by a Mann-Whitney test and now show box-whisker plots (see new Figures 1C, 4, 5C, and new Figure 2 —figure supplement 1, new Figure 4 —figure supplement 1, and new Figure 5 —figure supplement 1A and C).

In addition, ANOVA is used with different post-tests. Most of the time Dunn is used, but also Sidak and Tuckey are used. I could not infer from the dataset why that is and maybe the authors could explain.

As we now explain in the Methods, for parametric data we used one way ANOVA followed by a Tukey test; for nonparametric data, we used one way ANOVA followed by Dunn. When comparing at least three groups we used two-way ANOVA followed by Holm-Sidak, which has a better capacity than Tukey to discriminate potential significant differences. (It is more suitable when the p value is wanted, rather than confidence intervals.)

– In Figure 1C the significance was determined using a t-test? ANOVA seems more appropriate. Which statistical assay was used in Figure 3?

For greater statistical accuracy, we used all data points instead of the average of three regions of interest. We then used a non-parametric one-way ANOVA followed by a Dunn’s pot hoc test (new Figure 1C).

The test used in Figure 3 was described: “All results are expressed as mean ± SEM. Statistical significance was assessed in non-parametric Mann Whitney, one-sample t-tests using the indicated significance threshold (p)” When comparing each result (comparing the different conditions that correspond to single column) vs the value 1 (dotted line) we used one-sample t-tests. We removed the other statistical comparisons that were misleading because unnecessary.

– Figure 1A and B. Is the comparison between HTT-SD and HTT-SA significant?

Since we do not know the physiological level of phosphorylation in our experiment, we compared constitutive phosphorylation and absence of phosphorylation to the wild type condition. Consequently, we limited the number of post-hoc tests (to WT vs. HTT-SD and WT vs. HTT-SA only) to avoid loss of significance upon multiple comparisons. In our experimental conditions, the physiological level of HTT phosphorylation is rather low, which explains why we observe a significant difference in HTT-SD neurons but not in HTT-SA neurons. We mention this in the Discussion.

– Figure 2B. All the latency to fall of all the mice in all the repeat for all the days tested have been pulled together? If this is case, this representation of the rotarod data is unusual, please explain what the advantage is compared to just showing the curve of the entire experiment as done in panel C.

This is a valid point. We have modified the figure and show the curve of the entire experiment as done in panel C. We made the same modifications for the new Figure 8B, C.

– Figure 3C. The quantification has an abundance of asterisks. It is unclear what was compared with what.

Our intent was to show all the statistical differences but the asterisks did overburden the figure.

We now provide a simpler representation of the data with an emphasis on the key results: comparisons of each condition to the value 1.

– Fig S2. Why are HTT-SD and HTT-SA separately compared to the wild type control? They should be grouped together in the same chart and ANOVA should be used.

We separated the two genotypes because we compared each of them to their respective WT littermates, which were obtained from separate breedings (*Htt*^S421D/+^ x *Htt*^S421D/+^ crosses on one hand and *Htt*^S421A/+^ x *Htt*^S421A/+^ on the other). Consequently, the WT mice originate from different strains and cannot be grouped.

7. Why the authors do not elaborate more on the difference or lack thereof between HTT-SD and HTT-SA and they are so quick to drop the HTT-SA mutant from the rest of study? The alanine mutant are expected to be non phosphorylatable and thus should present a more drastic difference in relation to the constitutive phosphorylated mutant one (SD). The wild type HTT, which can still be phosphorylated, should be somewhere in the middle. This seems counterintuitive.

We initially focused on the HTT-SD mice because the effects on axonal transport of vesicles, their release at the synapse, and behavior were clearer. We have now added more data on HTT-SA mice. These new experiments indicate subtle alterations in the HTT-SA condition, which, although less marked, are the opposite of the alterations observed when HTT is consistently phosphorylated. We mention these points in the Discussion.

Have the authors tested for the phosphorylation status of their mutants?

When S421 is mutated into aspartic acid (mimicking constitutive phosphorylation) or into unphosphorylatable alanine, HTT is no longer recognized by anti-S421 phospho-HTT antibodies. See (Ehinger et al., 2020; Humbert et al., 2002) and new Supplementary Figure S5B.

8. It seems counterintuitive that HTT-SD promotes SVPs transport, but results in a behavioral deficit. Although the authors provide the explanation of these results, but it would be helpful if the authors could provide more evidence. For example, the hypothesis of an increase in docked SVs could be tested. The authors provide already EM images (Figure 4 and 7B). Could they quantify the number of docked vesicles? If this is difficult, maybe the number of vesicles adjacent to the membrane?

We were not able to quantify the number of docked vesicles but, as noted under points 1 and 4, we counted the number of vesicles at different distances from the membrane (new Figure 4C and 4D and new Figure 4 —figure supplement 1A). We found no difference in terms of the density of vesicles within the zone adjacent to the membrane in HTT-SD axon terminals. These results suggest that what changes in HTT-SD mice is not the number of docked vesicles at the presynapse but rather the density of the pool of vesicles distal from the membrane that is increased. While we cannot confirm the identity of this pool—location relative to the membrane does not constitute a reliable technique to define vesicular pool identities (Denker et al., 2009; Fernandez-Alfonso and Ryan, 2008; Fredj and Burrone, 2009)—these results are in agreement with the increased anterograde axonal transport of SVP we observed in HTT-SD neurons in vitro.

These findings are introduced in the Results section lines 183-204 and discussed in lines 377397.

References

Bodaleo, F. J., and Gonzalez-Billault, C. (2016). The Presynaptic Microtubule Cytoskeleton in Physiological and Pathological Conditions: Lessons from *Drosophila* Fragile X Syndrome and Hereditary Spastic Paraplegias. *Front Mol Neurosci, 9*, 60. doi:10.3389/fnmol.2016.00060

Bruyere, J., Abada, Y. S., Vitet, H., Fontaine, G., Deloulme, J. C., Ces, A.,... Saudou, F. (2020). Presynaptic APP levels and synaptic homeostasis are regulated by Akt phosphorylation of huntingtin. *ELife, 9*. doi:10.7554/*eLife*.56371

Chamberland, S., and Toth, K. (2016). Functionally heterogeneous synaptic vesicle pools support diverse synaptic signalling. *J Physiol, 594*(4), 825-835. doi:10.1113/JP270194

Chanaday, N. L., Cousin, M. A., Milosevic, I., Watanabe, S., and Morgan, J. R. (2019). The Synaptic Vesicle Cycle Revisited: New Insights into the Modes and Mechanisms. *J Neurosci, 39*(42), 82098216. doi:10.1523/JNEUROSCI.1158-19.2019

Colin, E., Zala, D., Liot, G., Rangone, H., Borrell-Pages, M., Li, X. J.,... Humbert, S. (2008). Huntingtin phosphorylation acts as a molecular switch for anterograde/retrograde transport in neurons. *Embo J, 27*(15), 2124-2134. doi:doi: 10.1038/emboj.2008.133.

Corradi, A., Zanardi, A., Giacomini, C., Onofri, F., Valtorta, F., Zoli, M., and Benfenati, F. (2008). Synapsin-I- and synapsin-II-null mice display an increased age-dependent cognitive impairment. J Cell Sci, 121(Pt 18), 3042-3051. doi:10.1242/jcs.035063

De Rossi, P., Nomura, T., Andrew, R. J., Masse, N. Y., Sampathkumar, V., Musial, T. F.,...

Thinakaran, G. (2020). Neuronal BIN1 Regulates Presynaptic Neurotransmitter Release and Memory Consolidation. *Cell Rep, 30*(10), 3520-3535 e3527. doi:10.1016/j.celrep.2020.02.026

Denker, A., Krohnert, K., and Rizzoli, S. O. (2009). Revisiting synaptic vesicle pool localization in the *Drosophila* neuromuscular junction. *J Physiol, 587*(Pt 12), 2919-2926. doi:10.1113/jphysiol.2009.170985

Ehinger, Y., Bruyere, J., Panayotis, N., Abada, Y. S., Borloz, E., Matagne, V.,... Roux, J. C. (2020). Huntingtin phosphorylation governs BDNF homeostasis and improves the phenotype of Mecp2 knockout mice. *EMBO Mol Med*, e10889. doi:10.15252/emmm.201910889

Fernandez-Alfonso, T., and Ryan, T. A. (2008). A heterogeneous "resting" pool of synaptic vesicles that is dynamically interchanged across boutons in mammalian CNS synapses. *Brain Cell Biol, 36*(14), 87-100. doi:10.1007/s11068-008-9030-y

Fredj, N. B., and Burrone, J. (2009). A resting pool of vesicles is responsible for spontaneous vesicle fusion at the synapse. *Nat Neurosci, 12*(6), 751-758. doi:10.1038/nn.2317

Gitler, D., Cheng, Q., Greengard, P., and Augustine, G. J. (2008). Synapsin IIa controls the reserve pool of glutamatergic synaptic vesicles. *J Neurosci, 28*(43), 10835-10843. doi:10.1523/JNEUROSCI.0924-08.2008

Goubard, V., Fino, E., and Venance, L. (2011). Contribution of astrocytic glutamate and GABA uptake to corticostriatal information processing. *J Physiol, 589*(Pt 9), 2301-2319. doi:10.1113/jphysiol.2010.203125

Humbert, S., Bryson, E. A., Cordelieres, F. P., Connors, N. C., Datta, S. R., Finkbeiner, S.,... Saudou, F. (2002). The IGF-1/Akt pathway is neuroprotective in Huntington's disease and involves Huntingtin phosphorylation by Akt. *Dev Cell, 2*(6), 831-837. doi:10.1016/s15345807(02)00188-0.

Kidokoro, Y., Kuromi, H., Delgado, R., Maureira, C., Oliva, C., and Labarca, P. (2004). Synaptic vesicle pools and plasticity of synaptic transmission at the *Drosophila* synapse. *Brain Res Brain Res Rev, 47*(1-3), 18-32. doi:10.1016/j.brainresrev.2004.05.004

Ratnayaka, A., Marra, V., Bush, D., Burden, J. J., Branco, T., and Staras, K. (2012). Recruitment of resting vesicles into recycling pools supports NMDA receptor-dependent synaptic potentiation in cultured hippocampal neurons. *J Physiol, 590*(7), 1585-1597. doi:10.1113/jphysiol.2011.226688 Rizzoli, S. O. (2014). Synaptic vesicle recycling: steps and principles. *EMBO J, 33*(8), 788-822. doi:10.1002/embj.201386357

Skorobogatko, Y., Landicho, A., Chalkley, R. J., Kossenkov, A. V., Gallo, G., and Vosseller, K. (2014). O-linked β-N-acetylglucosamine (O-GlcNAc) site thr-87 regulates synapsin I localization to synapses and size of the reserve pool of synaptic vesicles. *J Biol Chem, 289*(6), 3602-3612. doi:10.1074/jbc.M113.512814

Truckenbrodt, S., Viplav, A., Jahne, S., Vogts, A., Denker, A., Wildhagen, H.,... Rizzoli, S. O. (2018). Newly produced synaptic vesicle proteins are preferentially used in synaptic transmission. *EMBO J, 37*(15). doi:10.15252/embj.201798044